# Exploring New Ways to Analyze Data on the Spontaneous Physical Activity of Rodents Through a Weighing Balance

**DOI:** 10.3390/s25113290

**Published:** 2025-05-23

**Authors:** Pedro Paulo Menezes Scariot, Ivan Gustavo Masselli dos Reis, Walter Aparecido Pimentel Monteiro, Maria Clara dos Reis, Vanessa Bertolucci, Fulvia Barros Manchado-Gobatto, Claudio Alexandre Gobatto, Leonardo Henrique Dalcheco Messias

**Affiliations:** 1Research Group on Technology Applied to Exercise Physiology—GTAFE, Health Sciences Postgraduate Program, São Francisco University (USF), Bragança Paulista 12916-900, SP, Brazil; ivan.reis@usf.edu.br (I.G.M.d.R.); walter.monteiro@usf.edu.br (W.A.P.M.); maria.clara.reis@mail.usf.edu.br (M.C.d.R.); vanessa.bertolucci@mail.usf.edu.br (V.B.); leonardo.messias@usf.edu.br (L.H.D.M.); 2Laboratory of Applied Sport Physiology, School of Applied Sciences, University of Campinas, (FCA/UNICAMP), Campinas 13083-970, SP, Brazil; fgobatto@unicamp.br (F.B.M.-G.); cgobatto@unicamp.br (C.A.G.)

**Keywords:** spontaneous physical activity, rodents, load cell, weighing balance

## Abstract

**Highlights:**

**What are the main findings?**
We showed that it is possible to develop a cost-effective and robust SPA measurement system using user-friendly Arduino-based instrumentation.We analyzed SPA data in ways never explored before, adapting mathematical strategies, exploring SPA on a minute-by-minute basis, and classifying it into four distinct domains.

**What are the implications of the main finding?**
By offering a measurement method based on accessible instrumentation, we are contributing to the advancement of SPA-related research.Our expectation is that constant SPA monitoring could become a standard practice in both scientific research and veterinary settings.

**Abstract:**

**Background**: Weight-based methods can be cost-effective and practical for measuring spontaneous physical activity (SPA) in laboratory animals, but their adoption and exploration of analyses remain limited. **Methods**: We demonstrate the construction of a balance using accessible components (iron plates and open-source Arduino^®^ electronics) and provide detailed instructions to enable others to build their own systems. Additionally, we propose new analytical strategies, such as using the Mean of Weight Changes (MWC), assessing the dispersion of weight changes, and classifying SPA into domains, to enhance data interpretation. **Results:** The construction of the weighing balance using accessible components proved to be feasible, and the balance demonstrated sensitivity in distinguishing high SPA under experimental conditions known to modulate it (dark/light phases and small vs. large cages). Regarding the analyses, we were able to confirm that MWC analysis is a valid measure of SPA. Furthermore, the coefficient of variation in weight changes could be used as a complementary analysis to MWC. The proposed SPA domains also proved to be valid, as they aligned with the understanding that rodents spend a greater proportion of time in the higher SPA domains during the dark phase, while lower SPA domains predominate during the light phase. **Conclusions**: Our findings reinforce the robustness and validity of our weighing balance, designed using a low-cost setup based on iron plates and open-source Arduino^®^ electronics.

## 1. Introduction

Research with laboratory rodents plays a key role in advancing knowledge in the fields of biology. It is important to know the activities and behaviors of these animals in their cages, since they spend most of their time there. Although rodents’ physical activities have been referred to by different names in the past [1,2], they are now commonly recognized as spontaneous physical activity (SPA) [3]. Unlike deliberate exercise [3], SPA encompasses a range of movements performed throughout the day, such as fidgeting, grooming, rearing, postural maintenance, foraging, and ambulatory locomotion, which collectively contribute approximately 30% to daily energy expenditure [3]. Reports demonstrate that increased SPA is associated with improved physical fitness and reduced body adiposity [4,5,6,7,8,9,10,11,12,13,14].

Given its importance in the context of metabolism, SPA requires accessible measurement methods. In this context, the weight-based method provides a cost-effective and interesting alternative because it is based on a simple principle: movements, whether involving displacement or not, are quantified as weight changes (WC). To the best of our knowledge, weight-based methods for evaluating SPA in animals have been in use since the 1961 development of Cho’s apparatus [15], which assessed animal movement using a kymograph lever that was connected to a needle on an analog balance. Unlike analog predecessors, modern weighing systems are significantly more sensitive and better equipped to capture large amounts of data at higher sampling frequencies over extended periods (due to advancements in the manufacturing of instrumentation, including improved shielding for load cells). Due to advancements in weighing systems, research on SPA in laboratory animals under diverse conditions has significantly progressed. Studies have measured SPA (via WC) in investigations involving housing conditions [16,17], dietary modifications [2,18,19,20], exercise [21,22,23,24,25,26], diseases [27], and pharmacological effects [28]. Although all of these studies have significantly contributed to the expansion of knowledge, SPA data could be further explored to gain deeper insights.

Our main objective was to explore new ways for analyzing the SPA obtained from the weighing method. For this purpose, we modified a weighing system previously developed by our group [16,17,22] to a cost-effective setting, utilizing user-friendly Arduino-based instrumentation. Consequently, the mathematical strategy proposed by Biesiadecki, et al. [29] was adapted to our context, considering not only the mean, but also the dispersion, of WC. We also propose an analysis that classifies SPA values into different domains using quartile determinations. By segmenting SPA values into broader domains (for example, very low, low, moderate, and high), we believe it is possible to better categorize the quantity of SPA, thus facilitating the identification of movement patterns. In addition to defining SPA domains, we also analyzed the duration (in minutes) that animals spent in each domain, presenting multiple visualization formats, including segments by the hour, within an entire day, and over the full duration of the experiment. Proposing the analyses mentioned earlier would not be enough; they must be tested in animal models (a real-world application), particularly under conditions in which the results are predictable. Since rodents are typically more active during the dark phase than during the light phase [30,31], and larger cages are known to promote SPA [16,22,32], we conducted experiments comparing phases (dark vs. light) and cage sizes (small vs. large). This design was deliberately crafted not only to confirm expected outcomes, but, more importantly, to demonstrate the construct validity of our measurement system, thus also reinforcing the consistency of our analytical strategies.

Apart from the context of analysis, another relevant aspect to mention is the dissemination of SPA, which, unfortunately, remains insufficiently measured in the scientific context. This is partly due to a lack of awareness about the SPA concept, but mostly because a portion of researchers and professionals working with laboratory animals believe that only sophisticated and expensive technologies, such as infrared photo-beam systems or video tracking systems [33,34], are legitimate ways to measure SPA. This generates barriers for laboratories with limited resources, potentially slowing the progress of SPA-related research. For these reasons, we deliberately provided a detailed description of our construction process in the methods section. We hope that, by doing so, we can guide others who may wish to build their own reliable, simple, and low-cost weighing system by using accessible components (iron plates and Arduino^®^, an open-source electronics platform).

This manuscript is organized to guide the reader through the development and analysis of the present study. It begins with a description of the construction and development of the weighing balance, followed by an explanation of the load cell arrangement and signal acquisition system. Next, we discuss how raw data are transformed into weight values through calibration and outline the experimental setup, including the rodents and SPA recordings. The methodology section continues with an introduction to the algorithm used to analyze SPA and the mathematical strategies applied, including Biesiadecki’s summation method and the Mean of Weight Changes (MWC) approach. We also explore the concept of signal dispersion and its role in SPA analysis. The classification of SPA into distinct domains is explained, along with the creation of heat maps for minute-by-minute SPA domains. In the results section, we compare different mathematical strategies, present an overview of the 23-day experiment, and examine analyses across light and dark phases, as well as across different times of the day. Further analyses explore the SPA classification into domains, both within individual hours and across all experimental days.

## 2. Materials and Methods

### 2.1. Construction and Development of the Weighing Balance

The weighing balance was composed of two metal plates, with load cells fixed between them, as shown in Figure 1. Metal supports (MSs), with a thickness of around 4.75 mm, were cut so that they could be placed between the load cells and the plates. This is essential to prevent unwanted parts of the load cells from making contact with the plates. Only the MSs (where the screw mounting points were located) should be in contact with the plates, as visualized in Figure 1. After drilling the screw holes, MSs were affixed using super glue. We opted for bonding instead of welding the MSs because our plates deformed under the high temperatures of stick welding. Through prototype testing, we found that thicker plates could withstand the heat from welding; however, this would result in excessive weight on the load cells, potentially damaging them. Ultimately, we chose a plate thickness of 1.5 mm. Our weighing balance measures 60 cm in length and 47 cm in width (Figure 2), providing enough surface area to accommodate both small and large cages, as will be demonstrated later. While the upper metal plate (UMP) was cut steel, with a thickness of only 1.5 mm, the bottom metal plate (BMP) was designed like a tray, with the metal bend directed downward to create a contact-free area. The BMP was intentionally designed to accommodate screws and nuts for securing the load cells. Furthermore, all edges of the BMP in contact with the table generated a uniform weight distribution, while also minimizing vibrations. In summary, our weighing balance was designed to ensure consistent and accurate measurements.

### 2.2. Arrangement of Load Cells

The balance design, with load cells arranged in a triangular layout (Figure 2), was chosen due to the inherent geometric stability of the triangle, which ensures a more homogeneous distribution of weight across all load cells. This arrangement has been shown to be effective in previous studies that also assessed SPA through a weight-based method [16,17,22]. Regarding the positioning, we aimed to orient the triangle horizontally, with the apex (LC3) positioned along the shorter side of the plate (47 cm). We avoided placing the apex (LC3) along the longer side (60 cm), since significant vibrations at the edges of the plate could occur. Therefore, LC3 was positioned 23.5 cm from the edge of the plate, while the other two load cells (LC1 and LC2) were symmetrically placed, each 49 cm from the opposite edge. The load cells were spread out to minimize vibrations and provide stability.

### 2.3. Signal Acquisition System

In summary, amplified electrical signals from the load cells, acquired by the Arduino microcontroller, were converted to digital data and transmitted via a universal serial bus interface to a personal computer, where they were recorded for posterior analysis. An electrical diagram illustrating the connections is shown in Figure 3. We used three load cells of similar design (indiPESO^®^), each with a full-scale capacity of 5 kg and a nominal sensitivity of 2 mV/V. In total, to construct two weighing balances, we used two Arduino Nano^®^ (Arduino S.r.l., Monza, Italy) and six HX711^®^ amplifiers (Avia Semiconductor, Xiamen, China), which together, cost around USD 50. Regarding the wires of the load cell, red (E+), black (E-), green (A+) and white (A-) were connected to the HX711^®^, which is a 24-bit analog-to-digital converter. For each weighing balance, three load cells were connected to three HX711^®^ amplifiers, which were then linked to a single Arduino. The DT and SCK outputs of the HX711^®^ amplifiers were connected to the digital pins of the Arduino Nano^®^. Since each Arduino provides a single 5 V supply pin, we used a small breadboard to power the three HX711^®^ amplifiers.

### 2.4. Transforming Raw Data into Weight Values Through the Calibration of Load Cells

Since our weighing balance consists of three load cells, we conducted a three-step calibration process, with individually calibrated load cells. Different known weights (ranging from 1.8 g to 215 g) were placed on the UMP, positioned exactly at the screw mounting points of each LC. We recorded electrical signals for each known weight individually, for about 10 s. This calibration process is primarily essential for generating specific linear equations for each load cell (Figure 4a), enabling the conversion of electrical signals (denoted as x) into weight values (denoted as y, in grams), as expressed by Equation (1).(1)y=ax+b

Data conversion processes had to be carried out, as the balance did not provide ready-to-use values in weight units. In addition to converting the signal to weight, the calibration process is crucial for assessing linearity and sensitivity through the coefficient of determination (R^2^). Evaluating this coefficient is crucial because low R^2^ values may indicate electrical or mechanical defects in the load cell or problems with the balance system design.

Although our calibration system places the weights exactly at the screw mounting points of each load cell, it is important to highlight that weight can be measured from any point on the plate. To illustrate this, we designed an example (Figure 4b) considering that a 60 g lead weight is placed in the top left corner. Since each load cell is at a different distance from the 60 g weight, it is coherent that the three load cells do not perceive the 60 g weight equally. For example, LC3 is the most sensitive, due to its proximity to the 60 g lead weight, whereas LC1, positioned on the opposite side, registers negative values because the plate in that region experiences an upward tensile force. Ultimately, considering the characteristics of our weighing balance, it is important to remember that, to obtain the total weight (resultant), it is necessary to sum the weight values from all three load cells, as exemplified in Figure 4. The Standard Error of the Estimate (SEE) was calculated using the calibration data from the load cells. It was determined by taking the root mean value of the residual deviations between the observed and predicted measurements, adjusted for the degrees of freedom. The analysis yielded an SEE of approximately 0.4 g across the three load cells, indicating that the predicted masses deviate from the actual known values by only 0.4 g. This low level of error confirms that the system has satisfactory sensitivity to detect subtle weight variations in small animal models.

### 2.5. Rodents, Experimental Conditions, and SPA Recordings

A total of 20 recently weaned male C57BL/6J mice were obtained from Anilab^®^ in Paulínia, Brazil. Mice were randomly divided into two types of housing conditions: a small cage (SCage, n = 10) or a large cage (LCage, n = 10). During the experimental period, the mice were maintained under conventional sanitary conditions and fed a standard diet. All cages were made of polyethylene. As illustrated in Figure 5, mice kept in the SCage had a floor area of 495.3 cm^2^ (dimensions: length: 28.3 cm, width: 17.5 cm, and height: 16 cm). We used large cages with internal dimensions of 57 × 47 × 22 cm, providing a floor area of 2.679 cm^2^. The LCage (Model: BXP113 Z/I) was purchased from BioXP^®^, Produtos Para Biotério& Lab (SP, Brazil), a company that also manufactures specialized lids (mesh 05) designed to prevent mice from escaping. The mice were collectively housed at a density of ten animals *per* cage (49.5 and 267.9 cm^2^
*per* animal for the SCage and LCage, respectively). These same housing conditions were maintained during SPA recordings. Recordings were conducted over a one-month period, but only 23 days were deemed valid for analysis, as the remaining days were excluded due to power outages or unforeseen computer errors.

As already mentioned, recording the WC caused by the mice’s movements is the basis for obtaining their SPA. To make this possible, we simply placed the rodents’ cage on the weighing balance (Figure 6a). With respect to the duration of SPA recordings (Figure 6b), it must be mentioned that SPA recording started at 11:00 and continued until 09:00 the following day, totaling 22 continuous hours (12 h of light period and 10 h of dark period). SPA recordings were paused from 09:00 to 11:00, since this period was dedicated to laboratory procedures. The animals were handled during the dark phase, with red lamps turned on only during the manipulations. We prioritized working with the animals during the dark phase, considering that handling during the light phase could be stressful and disruptive to the sleep of the rodents [35]. The animal facility was programmed with a 12:00 h light/dark inverted cycle (illumination from 19:00 h to 7:00 h). Therefore, SPA values from 11:00 to 18:59 and 07:00 to 8:59 (with lamps off) were considered for the dark phase, while SPA values from 19:00 to 06:59 (with lamps on) were considered for the light phase.

### 2.6. Algorithm for Analyzing SPA

The algorithm developed to record and calculate SPA in the present study is available at https://github.com/igmdr/spontaneousactivity. The applications for both data storage and analysis were custom algorithms, designed specifically for this study. JupyterLab 3.6.3 (https://jupyter.org/ accessed on 1 March 2025) and Python 3.13.2 (https://www.python.org/ accessed on 1 March 2025), were used as the developmental environment and programming language, respectively. For the application that stores the signal, the pySerial 2.7 (https://pyserial.readthedocs.io/ accessed on 1 March 2025) library was used for signal acquisition, along with built-in Python functions to write data-to-text files. The Pandas library 2.2.3 (https://pandas.pydata.org/ accessed on 1 March 2025) was used for data import, data frame management, and performing arithmetic operations between matrices and arrays (such as computing mean, sum, standard deviation, and the difference between consecutive samples). The weight data were processed through a filter that removed outliers above and below two times the standard deviation of the respective period. For each balance, raw data were recorded in four columns, one for each of the three load cell signals and one for the time (in microseconds). To improve the accuracy and precision of the systems, we adopted the following strategy: We assumed that the minute with the lowest SPA within a day reflects the time when the animals exhibited minimal movement. Therefore, the SPA recorded during that minute likely represents the system’s baseline bias for that day. Consequently, we subtracted the value of this minimum activity from all minute-by-minute SPA values recorded on the same day. This correction enhances the accuracy of each system and reduces discrepancies between systems across the study period. Furthermore, it improves the consistency of measurements within each system across different days.

### 2.7. Mathematical Basis for Determining SPA: Biesiadecki’s Summation Strategy

Although obtaining weight is an inherent function of the balance, the primary objective is not merely to capture a single weight value (W), but to calculate weight changes (WC). We will now outline the necessary steps to understand this. The study by Biesiadecki, et al. [29] may be considered, to the best of our knowledge, the main work that focused on the mathematical basis for obtaining SPA. These authors calculated the WC between consecutive measurements, and then converted these values to their modulus (|⋅|). This transformation ensures that all values are non-negative, removing directional information, as downward movements are initially recorded as positive and upward movements as negative. In the final step of Biesiadecki’s strategy (SWC), SPA is obtained by summing the weight changes within each minute, as mathematically expressed in Equation (2).(2)$$SWC=\sum_{n=1}^{N}\left|w_{n+1}\right.-\left.w_{n}\right|$$

### 2.8. Mathematical Basis for Determining SPA: Mean of Weight Changes (MWC)

Despite its advantages, such as ease of operation and low cost, Arduino-based instrumentation exhibits inconsistencies in sampling rate. We modified the HX711^®^ amplifiers to capture signals at the highest possible frequency, which, in our case, was approximately 40 Hz. However, at certain moments, signal sampling exhibited fluctuations, leading to an increase in the number of samples per minute, which could result in an overestimation of SPA when using the SWC strategy. To solve this issue, we proposed the Mean of Weight Change (MWC) strategy, which introduces adjustments to account for variations in the sampling rate. As expressed in Equation (3), the MWC strategy also sums weight changes in modulus, following Biesiadecki’s principle, but with the distinction that the total WC is divided by the number of data points (N) collected per minute. In both mathematical strategies, the data from the three load cells are summed.(3)$$MWC=\frac{1}{N}\sum_{n=1}^{N}\left|w_{n+1}\right.-\left.w_{n}\right|$$

### 2.9. Exploring Beyond MWC: A Look at Signal Dispersion

MWC itself is a robust measure of SPA; however, we also contemplated that there may be situations in which certain nuances or details might not be apparent when looking at MWC alone. Specifically, we calculated the standard deviation (SD) and coefficient of variation (CV) to assess the dispersion of weight changes. As illustrated in Figure 7, hypothetical examples were simulated, demonstrating that, while identical MWC (2.00) can occur under certain conditions, the CV can differ (86.6 vs. 143.4%), indicating distinct signal variation.

### 2.10. SPA Classification into Domains

For the entire dataset (across all days and both cage types), the values 11.0, 16.8, and 22.4 correspond to the 25th percentile (P25), 50th percentile (median, P50), and 75th percentile (P75), respectively, as shown in Figure 8. From these delimiters, a quartile distribution was generated. In an attempt to characterize a profile for each domain, we will speculate (in the discussion section) on possible behaviors/activities that could be found within the domains, taking into account the divergences between cages.

Using the defined quartiles (Qs), SPA was categorized into four distinct domains in Excel^®^ using Equation (4). This classification was applied to the SPA data (obtained through the MWC strategy) on a minute-by-minute basis.=IF(SPA < 11.0, “1”, IF(SPA < 16.8, “2”, IF(SPA ≤ 22.4, “3”, “4”)))(4)

To systematically categorize the SPA, the numerical values 1, 2, 3, and 4 were assigned to represent VL_SPA_D, L_SPA_D, M_SPA_D, and H_SPA_D, respectively. With this classification established, the *COUNTIF* function in Excel^®^ was used to determine how many minutes were spent in each domain. The time spent in each domain was presented in three ways: within a full hour (as illustrated in Figure 9), across an entire day, and over the course of the experiment (as illustrated in Figure 10).

### 2.11. Minute-by-Minute Heat Maps of SPA Domains

Due to the simplified classification system of 1, 2, 3, and 4, which represent the VL_SPA_D, L_SPA_D, M_SPA_D, and H_SPA_D, respectively, it was possible to assign each domain a distinct color. This enabled the creation of heat maps (Figure 10), which, due to their visual nature, are highly intuitive and facilitate a clearer interpretation of the data. This four-color classification not only provides an appealing way to present large datasets (in our case, minute-by-minute SPA data), but also allowed us to show how the SPA domains were distributed within an entire day and throughout the course of the experiment.

### 2.12. Statistical Procedures

All data were presented as mean ± standard deviation. GraphPad Prism software (version 8.2.1) was used to generate graphs, and STATISTICA (StatSoft, Inc. (Tulsa, OK, USA) 2004, version 7) was used for statistical analyses. All data were assessed for normality via a Shapiro–Wilk test. In the initial analysis, with the primary focus solely on comparing housing conditions without accounting for temporal dynamics, we employed an independent t-test to assess the difference between the SCage and LCage. To enable a more detailed exploration of SPA, we conducted a two-way ANOVA to evaluate the effects of housing conditions (SCage vs. LCage) and phase (light vs. dark). Finally, to achieve finer temporal resolution, we assessed hour-by-hour SPA fluctuations using an additional two-way ANOVA, with housing conditions and ’hour’ as factors. Finally, we conducted another two-way ANOVA to determine whether the time spent in each domain (VL_SPA_D, L_SPA_D, M_SPA_D, and H_SPA_D) varied, and whether this variation was influenced by housing conditions. Non-normal data were transformed using the inverse normal transformation in IBM SPSS Statistics Version 26 [36]. For cases in which the inverse normal transformation did not succeed in achieving normality (as with the results presented in Table 2), we employed a rank transformation approach [37]. To implement it, the entire set of observations was ranked (using IBM SPSS Statistics) from smallest to largest, i.e., assigning rank 1 to the smallest value, rank 2 to the next smallest, and so on, up to the maximum value, which received the highest rank. The ranked data were then analyzed using a traditional two-way factorial ANOVA. Tukey’s post hoc test was chosen for all situations. The significance level was set at *p* < 0.05 in all cases. The effect size (Cohen’s d) for pair-wise comparisons was determined. We applied the trapezoidal method [38] to determine the total area under the curve (AUC) from the SPA measurements obtained over the entire 23-day period.

## 3. Results

### 3.1. Comparative Analysis of Mathematical Strategies (SWC vs. MWC)

Rodents housed in a large cage exhibited greater SPA than those housed in a small cage, and this occurred in both mathematical strategies (Figure 11). We also observed strong and significant correlations (r = 0.99) between the mathematical strategies within each cage type.

### 3.2. A Panoramic View on the 23-Day Experiment

Since the MWC has proven to be a valid mathematical strategy, we will now proceed with data analysis. By using a single daily SPA value, it is possible to present an overview of the data collected throughout the experiment (23 days, in this case), as shown in Figure 12a. Using an independent t-test, we observed that daily SPA was higher in the LCage compared to the SCage (Figure 12b). In accordance with these statistical findings, the cumulative SPA throughout the entire experiment can be visualized by calculating the area under the curve (AUC) over the 23 days. Although the AUC reveals that the SPA in the LCage is 12% higher than that in the SCage, it should be noted that the AUC itself does not permit statistical analysis, as it represents a single area value.

### 3.3. Exploring Analyses Across Light and Dark Phases

Figure 13 illustrates the analysis of SPA (MWC) behavior across different phases (dark vs. light) under different housing conditions. ANOVA revealed a significant effect of housing, with mice in the LCage exhibiting higher SPA than those in the SCage. However, post hoc analysis revealed that the increased SPA in the LCage was only significant during the dark phase (*p* < 0.001). In contrast, no significant differences were observed between the SCage and LCage during the light phase (*p* = 0.091). These findings help explain the significant interaction detected by the two-way ANOVA. The effect of housing on SPA is phase-dependent, meaning that the large cage boosts SPA exclusively during the dark phase, with no impact during the light phase. Regarding phase comparisons, we found a significant main effect of phase, indicating that SPA was higher during the dark phase compared to the light phase, and this result aligns with the post hoc analysis (Figure 13a).

### 3.4. Exploring Analyses Across Different Hours of the Day

When exploring SPA per hour, interesting interpretations can be extracted, as illustrated in Figure 14. In line with ANOVA (Figure 14a), the LCage exhibited higher SPA than the SCage only during the hours when the lights were off. Figure 14b shows detailed post hoc significant differences between the LCage and SCage. Higher SPA in the LCage was observed during specific time segments: from 11:00 to 14:59, 17:00 to 18:59, and 07:00 to 08:59. No significant differences were observed between the SCage and LCage during the hours within the light phase.

An effect of hour on SPA was found, showing that SPA fluctuates in a pattern of elevations and troughs throughout the day (F = 125.6, *p* < 0.001). This is expected for nocturnal rodents. A significant interaction between housing and hour on SPA was also observed, suggesting that the hour-of-day fluctuation of SPA differs between the SCage and LCage. Despite this, the hours with the highest and lowest levels of SPA occurred during the dark and light phases, respectively. Additional information on the exact times when SPA reached its minimum and maximum values can be found in Figure 15.

At what times did the lowest and highest SPA occur? This is a common question in the field of chronobiology. Our algorithm enables us to extract not only the hours, but also the exact minutes, of the nadir and acrophase of SPA, as illustrated in Figure 15. We also highlight, in yellow and gray, the times that correspond to the light phase (lamps on) and the dark phase (lamps off). Additionally, analyzing the highest SPA, we observed significant differences between the SCage and LCage (Cohen’s d = 2.62, t = 8.43, *p* < 0.05).

### 3.5. Signal Dispersion Analyses

Table 1 presents the standard deviation of weight changes (SD-WC) for the small and large cage across different phases (dark, light, and overall). However, we did not perform statistical analyses for SD, as we considered it more appropriate to compare CV. As a descriptive measure, the SD-WC varied across phases, with higher values observed during the dark phase. SD-WC was higher in the large cage (23.1), compared to the small cage (18.1), during this phase. Conversely, during the light phase, the SD-WC was lower in both groups, with values of 11.3 in the small cage and slightly lower in the large cage (10.1). When considering all phases combined, the SD-WC was 14.4 for the small cage and 16.0 for the large cage.

The coefficient of variation of weight changes (CV-WC) was analyzed under two housing conditions, exploring phases (Figure 16a) and hours (Figure 16b). The ANOVA revealed a significant effect of phase (F = 50.4, *p* < 0.001), showing that CV-WC during the light phase was higher than during the dark phase. However, interestingly, the CV-WC was significantly higher in the SCage compared to the LCage only during the light phase (but not during the dark phase), in accordance with a significant interaction found (F = 116.7, *p* < 0.001). During the dark phase, there was no difference between SCage and LCage (Cohen’s d = 0.05). ANOVA revealed that the CV-WC remained higher in the SCage than in the LCage during specific hours of the light phase (21:00 to 6:59), while no differences were observed during the dark phase (Figure 16b). This aligns with the significant interaction found (F = 1.07, *p* < 0.001).

### 3.6. SPA Classification into Domains Within Hours

Regarding comparisons with rank-transformed values, it was possible to perform an ANOVA (Table 2). A significant interaction was observed (*F* = 114.7, *p* < 0.001), indicating that the time spent in different SPA domains varied depending on the time of day. Specifically, this interaction is illustrated in the bar graphs (Figure 17), which show a clear predominance of H_SPA_D and M_SPA_D during the dark phase (lights off), while VL_SPA_D and L_SPA_D were more predominant during the light phase (Table 2). Regarding the interaction between housing and domain (*F* = 104.9, *p* < 0.001), ANOVA results indicate that H_SPA_D and VL_SPA_D were more pronounced in rodents housed in the large cage, whereas L_SPA_D and M_SPA_D were more prominent in those housed in the small cage. The distribution of time suggests that animals followed a circadian pattern of rest and activity, aligning with their biological nature.

### 3.7. SPA Classification into Domains Across All Experimental Days

Table 3 presents the descriptive results of the distribution of time (in minutes) spent in each SPA domain across all experimental days, considering only the dark phase. The ANOVA results (F = 347.5, *p* < 0.001) indicate a significant difference in the time spent across the four domains, with the following pattern: H_SPA_D > M_SPA_D > L_SPA_D > VL_SPA_D. A detailed examination of the post hoc analysis shows that, in the small cage group, M_SPA_D was predominant, accounting for the largest proportion of time (287 min on average) compared to VL_SPA_D (*p* < 0.001), L_SPA_D (*p* < 0.001) and H_SPA_D (*p* < 0.001). For the large cage group, H_SPA_D was predominant, accounting for the longest duration (375 min on average) compared to VL_SPA_D (*p* < 0.001), L_SPA_D (*p* < 0.001) and M_SPA_D (*p* < 0.001). Still, for the dark phase, a significant interaction between housing and domain (F = 93.2, *p* < 0.001) reinforces the idea that cage size may affect the distribution of time across SPA domains. No differences were observed between the SCage and LCage with respect to the VL_SPA_D and V_SPA_D.

Table 4 presents the descriptive results of the distribution of time (in minutes) spent in each SPA domain across all experimental days, considering only the light phase. The ANOVA results (F = 362.5, *p* < 0.001) indicate a significant difference in the time spent across the four domains during the light phase, with the following pattern: VL_SPA_D > L_SPA_D > M_SPA_D > H_SPA_D. Post hoc analysis revealed that, in the small cage group, no differences were found between the VL_SPA_D and V_SPA_D. Both VL_SPA_D and L_SPA_D were predominant, accounting for the largest proportions of time (282 and 263 min on average, respectively), compared to M_SPA_D and H_SPA_D (*p* < 0.001). For the large cage group, VL_SPA_D was predominant, accounting for the longest duration (358 min on average) compared to L_SPA_D (*p* < 0.001), M_SPA_D (*p* < 0.001), and H_SPA_D (*p* < 0.001). A significant interaction between housing and domain was found (F = 16.7, *p* < 0.001), indicating that the amount of time spent across SPA domains is dependent on housing. No differences were observed between the small cage and large cage groups with respect to the L_SPA_D and H_SPA_D.

### 3.8. SPA Classification into Domains Throughout the Experiment

Complementing Table 3 and Table 4, we present, in Figure 18, the summed minutes spent in each SPA domain across the entire experiment. During the dark phase, rodents in the large cage spent a larger proportion of time in the H_SPA_D (63%), while those in the small cage spent more time in the M_SPA_D (48%). In contrast, during the light phase, the majority of SPA occurred in the VL_SPA_D and L_SPA_D for small cage. Notably, rodents in the large cage spent 50% of their time in VL_SPA_D, compared to 39% in the small cage.

Figure 19 illustrates the heat maps showing the temporal distribution of SPA domains throughout the 23-day experiment for rodents housed in small and large cages. Activity levels varied across the light and dark phases, with higher SPA observed predominantly during the dark phase (lights off), as indicated by the darker regions. In both cage conditions, SPA was lower during the light phase (lights on), with predominance of VL_SPA_D and L_SPA_D.

## 4. Discussion

There is a growing demand for methods to measure SPA, especially those using more affordable and accessible resources, such as Arduino-based instrumentation. Even with a simple and low-cost system, it is possible to develop a reliable and consistent weighing system that is particularly sensitive to discerning SPA in response to environmental influences (e.g., cage size and light/dark phases). To contribute to the dissemination of SPA, we provide a detailed description of our construction process. Specifically, we contribute the following details: (1) how the iron plates were dimensioned, (2) how the load cells were arranged in a triangular layout, (3) illustration of the wiring diagram of the signal acquisition system, and (4) how the balances were calibrated. Although these details have been explored previously [16,17,22], several construction steps were revisited to clarify the modification proposed here, considering the low-cost perspective based on Arduino-based instrumentation. With our detailed textual instructions as well as real photos (provided in the Appendix A), we expect that anyone will be able to reproduce the weighing method.

We examined SPA data using strategies that had previously never been explored. Since differences between the SCage and LCage were detected in both the SWC and MWC strategies, we highlighted the relevance of both mathematical strategies. Despite the effectiveness of the SWC strategy, the use of MWC may have some benefits in cases where there are different sampling rates. In addition to this, we calculated the CV-WC as an additional analysis to further explore SPA. If we consider only the results from MWC, we would conclude that there are no differences between large and small cages on SPA (during the light phase). However, we observe a lower CV-WC in animals housed in the large cage during the light phase. This allows us to speculate that rodents in the LCage experience deeper rest during the light phase, which could explain their greater readiness during the dark phase.

The SPA data demonstrate the consistency of our weighing balance. For example, rodents are naturally more active in the dark phase, when compared to the light phase, in agreement with the literature [30,31]. The coherence of rodent chronobiology is also evident in our proposed SPA domains. There was a clear predominance of the H_SPA_D and M_SPA_D during the dark phase, while the L_SPA_D and VL_SPA_D predominate during the light phase. We also explored the data regarding the highest and lowest SPA values. This analysis was consistent with the biological nature of rodents, but also revealed that the times at which the highest and lowest SPA occurred were not always the same; in other words, rodents do not strictly follow a rigid schedule.

Regarding heat maps, it is notable to mention that they surprised us by revealing a high consistency, visually confirming the clear predominance of the H_SPA_D and M_SPA_D during the dark phase, while the L_SPA_D and VL_SPA_D predominated during the light phase. With careful consideration, our heat maps resemble a modified version of actograms, which are commonly used to illustrate biological rhythms [39,40,41,42,43]. Although our analysis was limited to visual inspection, we believe that heat maps can reveal patterns/trends that are not easily noticeable in tables. One example of this is the success of heat maps in synthesizing information on the four SPA domains throughout the minutes of a day and over multiple days. This contributes to the field of chronobiology of rodents. However, it is important to note that our intention is not to claim that our heat maps function as actograms. We did not explore variables typically derived from actograms, such as the endogenous period (τ), T-cycles, or Zeitgebers. It is important to mention that our experimental design was not perfect for analyzing rhythmic variations.

Although previously discussed in other studies [16,17,22], it is essential to reiterate that the choice to use three load cells instead of just one is entirely based on the load cells’ full-scale capacity. This capacity determines both of the following: (a) the maximum weight the load cells can safely measure without damage (safe overload) and (b) the sensitivity required for accurate SPA measurements. When considering the maximum weight that the balance should support, it is essential to account for not only the upper metal plate (UMP), but also the combined weight of the plastic cage and all of its contents (e.g., bedding, water bottle, food, and animals). Since our weighing balance was designed to accommodate cages of different sizes, we predicted a maximum weight of 10 kg. Thus, we could not use load cells with a full-scale capacity lower than or equal to 10 kg. While it may seem logical to use a 15 kg load cell, this comes with a disadvantage (a lower sensitivity for detecting lighter weights under 3 g, which are expected in the rodents’ movements). This led us to choose three 5 kg load cells instead of a single 15 kg load cell. Although a 5 kg load cell has a lower maximum capacity, multiple 5 kg load cells can be arranged together to improve weight distribution. Importantly, 5 kg load cells offer better resolution (smaller full-scale range), making them more suitable for detecting the sensitive movements of the animals.

As demonstrated earlier, we achieve an ideal balance between handling the expected weight load and maintaining the sensitivity required for accurate SPA measurements by using three 5 kg load cells. Although we believe that the sensitivity level in our study has been sufficient to register not only horizontal activities (ambulation), but also activities that occur without movement along the horizontal plane (e.g., fidgeting, grooming, and hearing), we recognize that the lack of video recordings may be a limitation of our study. Without a synchronized relationship between the weight measurements and visual recordings, we are unable to accurately determine which behaviors/activities were performed, especially those within each domain. Next, we will provide a didactic profile for each SPA domain, but we want to be cautious, as we are only speculating probable behaviors and characteristics.

***Very Low SPA Domain (Q1):*** *Minimal amount of SPA, predominantly occurring during some specific moments in the light phase. In this domain, rodents are expected to be mostly inactive, with periods predominantly spent quietly resting or sleeping. Movements are likely limited to brief shifts in position or grooming behaviors, with little to no sustained displacement.*

***Low SPA Domain (Q2):*** *This domain may be associated with quiescent behaviors and short locomotor displacements. Given that it is unlikely that rodents remain in a sleep-like state throughout the entire light phase, moderate exploratory behaviors (e.g., investigative sniffing, brief rearing) may occasionally occur within this domain.*

***Moderate SPA Domain (Q3):*** *For rodents in large cages, this domain appears to reflect a transition between phases, during which they engage in active preparatory behaviors such as nesting, burrowing, and rearranging bedding as they prepare for sleep, or exploratory movements as they gradually return to wakefulness during the dark phase. In contrast, for rodents in small cages, it seems that this domain is where the animals are most active. This domain predominantly occurs during the dark phase, and it appears that the animals are nearing their maximum limits for movement. Perhaps due to the inherent restrictions of the small cage, certain activities are likely limited compared to those of rodents in the large cage.*

***High SPA Domain (Q4):*** *This domain represents a substantial amount of SPA, predominantly occurring during the dark phase. It is possible that, in this domain, high-intensity physical activities such as fast running, exploration, and dominance-related behaviors (including territorial disputes and aggressive interactions) occur. This domain may also involve intense food-seeking behaviors and an increase in non-aggressive social interactions, as the rodents engage in social bonding or affiliative behaviors. As animals kept in the small cage had limited access to this domain, it is reasonable to assume that some of the behaviors described for this domain may be present (perhaps more suppressed) in the moderate domain.*

SPA domains were defined based on the amount of SPA, and it is important to note that this should not be confused with a measure of SPA intensity. Physiological markers (such as lactate or oxygen consumption), which are essential for defining intensity, are not available in our study, which could be considered a limitation. However, even if such markers were available, it is likely that the majority of SPA would fall into the light-intensity domain. This seems reasonable, since SPA encompasses a range of movements (such as fidgeting, grooming, rearing, and postural maintenance) that are unlikely to cause significant increases in lactate or oxygen consumption.

Although we are not validating our balance against an external reference method, we tested whether the system responds to independent experimental manipulations that, according to well-established theoretical frameworks, are known to influence SPA. Since the hypotheses that the dark phase and larger cages would result in higher SPA were not derived from the system’s data, but rather from findings in the literature, we assert that no circular analysis was conducted. In our view, this approach constitutes a form of construct validity.

Considering that we explicitly stated that SPA measurements were conducted in animals housed collectively, it is evident that the recorded WC may reflect the combined activities of all the animals, making it difficult to isolate the behavior of a single subject. While this is a point of critique for some researchers, we hold a different perspective. We believe that our collective-based measurements were well-founded, preserving social housing while accurately reflecting the rodents’ natural daily behaviors. Our view is in agreement with animal welfare regulations, which discourage the use of single housing for social species, such as C57BL/6 mice [44,45,46]. Many researchers obtain individual SPA values by isolating a single animal per cage, but this practice is highly detrimental. Animals removed from a collective environment and temporarily isolated in an unfamiliar setting for a few days may become anxious and stressed due to the separation from their cage mates [47,48]. Additionally, this temporary disruption of established social relationships (hierarchies) can cause significant stress and lead to an increase in aggressive interactions among the remaining animals in the cage [44]. It is important to recognize that SPA is not merely a measure of locomotion. Instead, SPA should be assessed in animals within their natural habitat (or home cage) and, ideally, through long-term monitoring that minimizes disturbances to their normal life and behavior. A potentially superior option is the use of bioimplants based on telemetry systems, such as those using accelerometers or radio-frequency-based tracking devices. However, although these devices can provide individualized data on animal movements, there are still limitations, including higher costs, invasiveness due to the need for surgical implantation, and the potential for interference if transmitters operate on overlapping frequencies [49].

Measuring SPA (through weighing balances) offers an accessible and non-invasive alternative, particularly when collective housing is required. Further, SPA monitoring can provide valuable insights for research related to health, behavior, and metabolism. Some examples of research areas in which SPA monitoring can be particularly valuable include dietary interventions (e.g., high-fat diets, food restriction), both acute and chronic exercise programs, as well as studies involving diseases, surgeries, injuries, or medications. A decrease in SPA may indicate a worsening clinical condition in animals, while an increase could suggest recovery and improved physical fitness. Monitoring and establishing reference SPA values for each animal facility could be integrated into the context of animal welfare. Any abnormality in SPA could serve as a potential indicator of stress, fear, or disruption in social relationships. Possible causes include errors in relocating a rodent to an unfamiliar cage, irregular food availability, poor bedding cleanliness, and the presence of intruder animals that may have entered through drains or unnoticed gaps. Additionally, SPA abnormalities could also result from human interference, including mistreatment by untrained or ill-intentioned animal handlers, or unauthorized entry into the animal room at inappropriate times, such as during the light phase, when animals should not be disturbed during sleep. All of these situations could be avoided with a diligent bioterium team conducting constant SPA monitoring. Due to its compact design and versatility, a weighing balance can be placed on any stable shelf or table within the animal facility, allowing SPA measurements to be conducted in the animals’ familiar environment, where they feel most comfortable and experience less stress. SPA data can be useful in determining whether animals are being maintained under optimal conditions. This not only promotes animal welfare, but also enhances good scientific practices.

Another key strength of this method is its ability to generate vast amounts of monitoring data. For instance, we were able to monitor approximately 506 h of data throughout the experiment (for 22 h per day over 23 days), demonstrating the extensive coverage of the exploration. This extended and continuous monitoring period provides experimental advantages, enabling bioterists and scientists to make more complete and accurate interpretations of animal behavior. Beyond the duration of monitoring, the sheer volume of data collected is also remarkable. For example, considering a sampling rate of 40 Hz, we captured approximately 72 million data points per load cell throughout the present study. Given that we used two balances, each equipped with three load cells, the total dataset amounted to approximately 437 million data points. This ensures a more detailed behavioral analysis. An important point to highlight is that weight-based SPA assessments still hold significant untapped potential. For example, future studies could estimate the animal’s position (center of mass) using load cell signals, thereby enriching SPA analysis with spatial information and enabling more refined assessments of movement trajectories and exploratory patterns within the cage.

This study is exploratory in nature and not without statistical and logistical limitations, including the absence of sample size calculations, the unavailability of additional balances to assess multiple cages simultaneously, and the fact that we tested the system using only one rodent species (C57BL/6 mice), which may limit the generalizability of the findings to other strains or species. Nevertheless, we sought to design an experiment that was grounded by previous research [16,17,21,22,23], which has consistently demonstrated reliable assessments of SPA in groups of 10 rodents per cage. Further, the experimental housing conditions were carefully designed to align with established guideline recommendations.

By providing detailed instructions and practical guidance, our intention is to empower others to build their own weighing balance, making weight-based SPA measurements a standard practice in both scientific research and veterinary settings. The information provided in this study is not only valuable in the scientific scope, but also relevant in education for young students who utilize programming languages (e.g., Python) or have related engineering disciplines in their curriculum. While we cannot directly measure its social impact, we hope that our contributions generate a positive effect on these students, nurturing their development for future technological and scientific challenges. Independent of commercial interests, professionals in the industry may also find value in our weighing balance. This, ultimately, opens opportunities for innovation and advancements in new weighing balance prototypes specifically designed for different situations and animal species, potentially expanding SPA measurement. In this regard, it would be desirable for SPA analysis not to be limited to rodents (such as rats, mice, and guinea pigs), but to be extended to a wider range of small animals such as rabbits, as well as reptile and amphibian species, commonly kept in captivity. Even larger balances could prove useful for assessing larger animals, such as horses, pigs, or even cattle, in confinement.

## Figures and Tables

**Figure 1 sensors-25-03290-f001:**
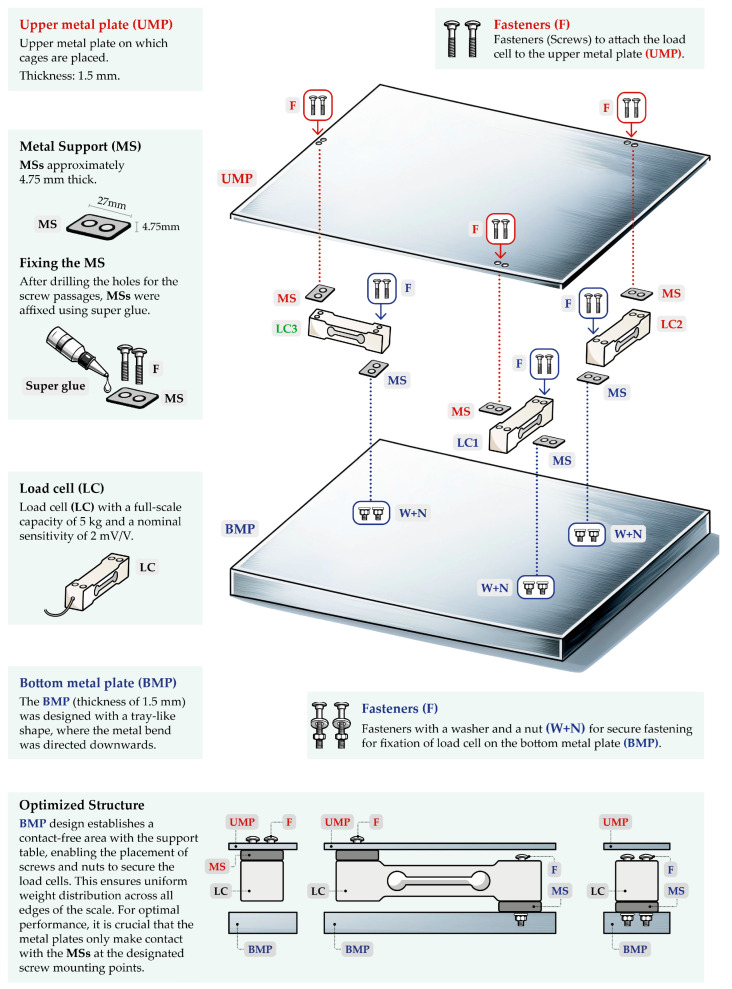
Weighing balance setup shows how each load cell (LC) was fixed between the plates. The upper metal plate (UMP) supports the weight of the animal cage, while the bottom metal plate (BMP) has downward-folded edges to form a tray. This design of the BMP allows space for fasteners (screws, nuts, and washers) to secure the load cells. The figure was created by one of the authors (V.B.) and is free of copyright restrictions. The figure elements are not to scale.

**Figure 2 sensors-25-03290-f002:**
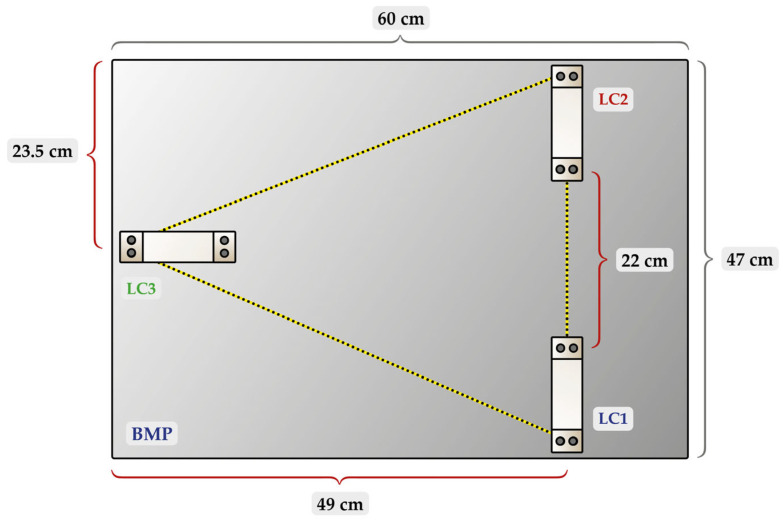
Illustration of the weighing balance from an overhead view, highlighting (dashed lines) the triangular arrangement of the load cells. The figure was created by one of the authors (V.B.) and is free of copyright restrictions.

**Figure 3 sensors-25-03290-f003:**
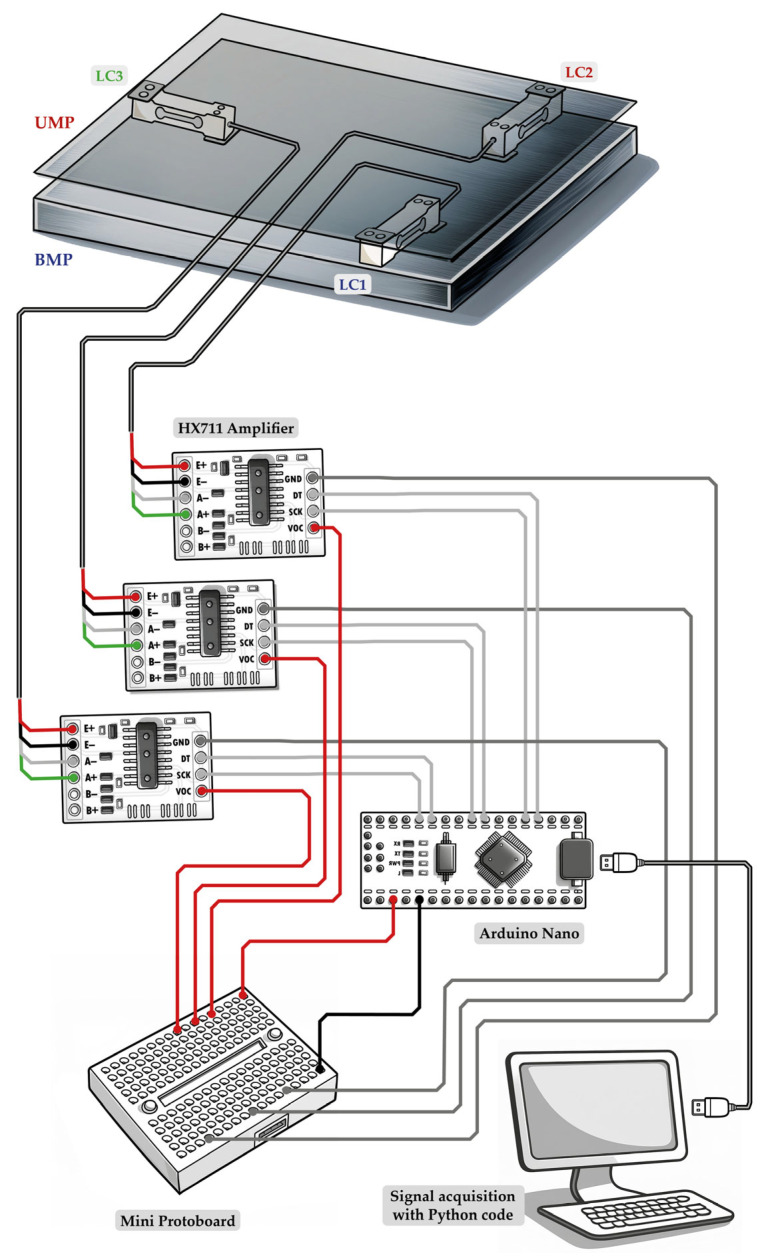
Wiring diagram of the signal acquisition system. The electronic components used were load cells, the Arduino Nano^®^, and the HX711^®^ amplifier. The figure was created by one of the authors (V.B.) and is free of copyright restrictions. The figure elements are not to scale.

**Figure 4 sensors-25-03290-f004:**
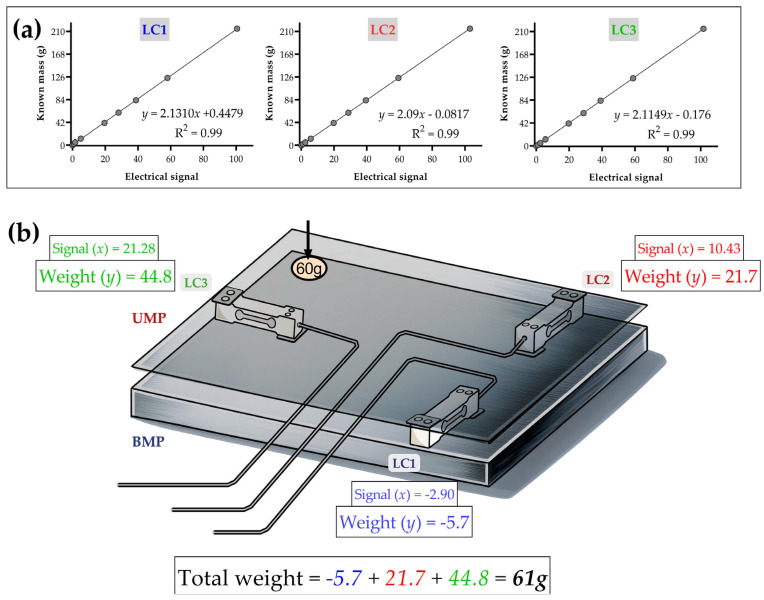
The calibration process involved applying different known weights directly to the mounting points of each load cell (LC) on the upper metal plate (UMP) to obtain linear regression equations (**a**), which will later be used for signal-to-weight conversion. If a known 60 g lead weight is placed in the top left corner, the load cells will perceive this weight differently, but, when summed, they will estimate the total weight applied to the balance (**b**). The figure was created by one of the authors (V.B.) and is free of copyright restrictions.

**Figure 5 sensors-25-03290-f005:**
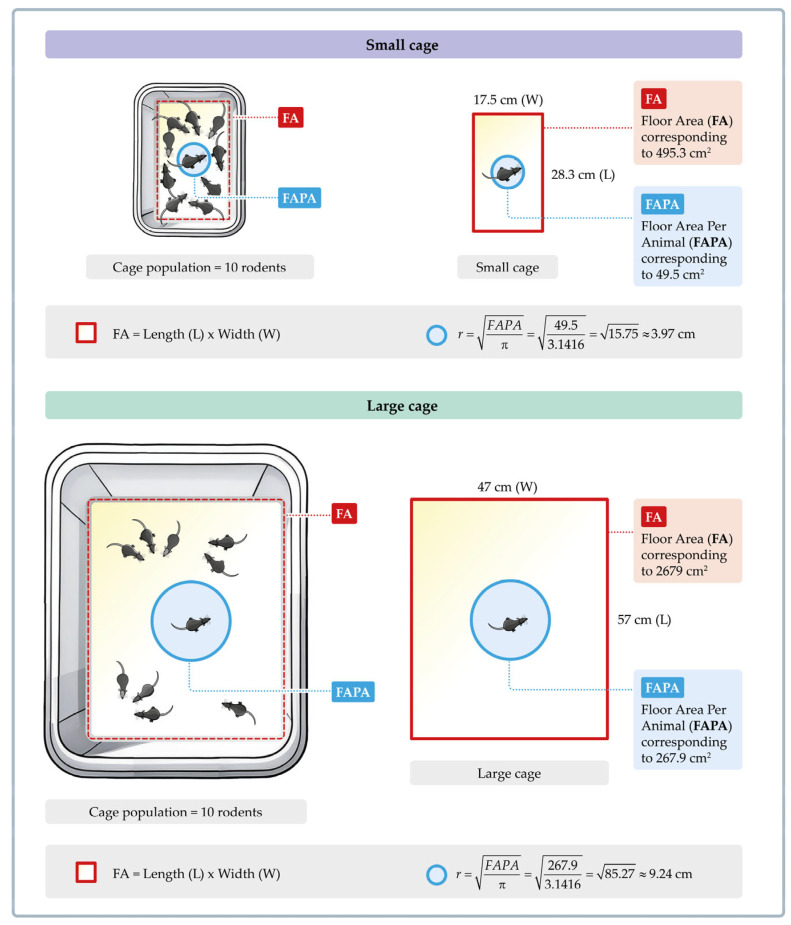
Comparison between small and large cage dimensions. The floor area per animal (FAPA) was calculated by dividing the total floor area (FA) of the cage by the number of rodents (n = 10). The small cage has a lower FAPA (49.5 cm^2^) compared to the large cage (267.9 cm^2^). This suggests that the smaller available space per rodent in the small cage may limit each animal’s potential radius of movement. This concept is illustrated by the blue circles, which represent the radius of available space for each mouse. In the small cage, this radius is 3.97 cm, while, in the large cage, it increases to 9.24 cm. The figure was created by one of the authors (V.B.) and is free of copyright restrictions.

**Figure 6 sensors-25-03290-f006:**
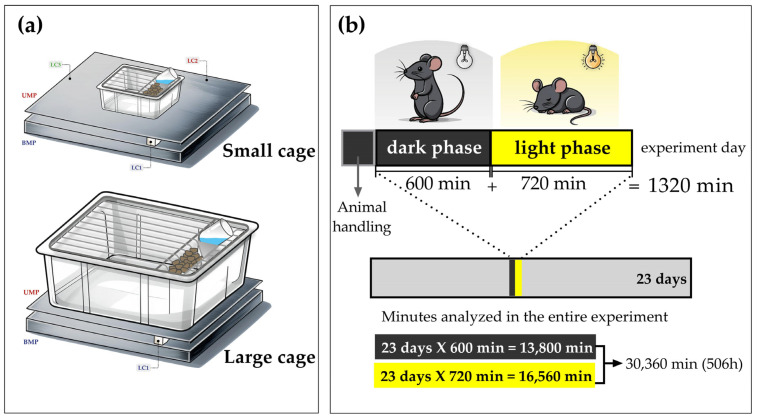
Rodents’ cages on the weighing balance (**a**), and the total number of minutes analyzed in the present study (**b**), distinguishing the dark phase (600 min/day) and the light phase (720 min/day). The total duration of the experiment was 23 days, summing up to 13,800 min in the dark phase and 16,560 min in the light phase, and totaling 30,360 min (506 h) of recorded activity.

**Figure 7 sensors-25-03290-f007:**
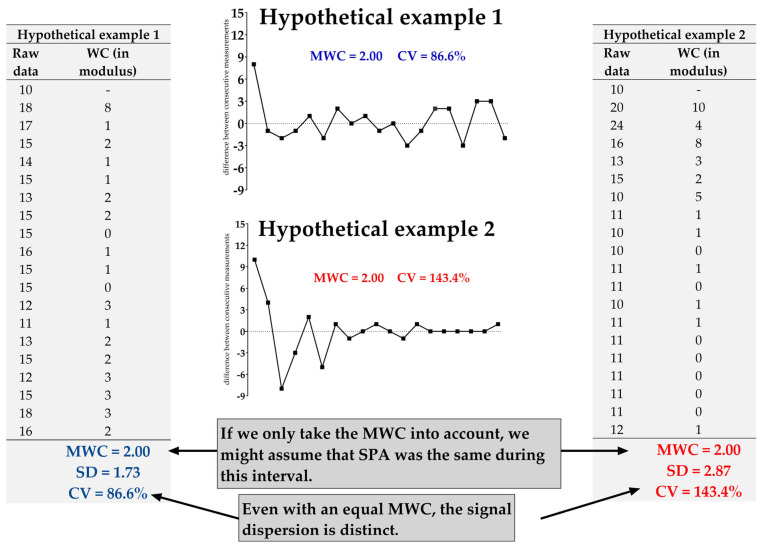
Fictional example illustrating that two signals with similar Mean of Weight Changes (MWC) can exhibit different dispersion due to differing coefficients of variation (CV). This demonstrates the importance of not focusing solely on MWC, but also exploring signal dynamics to avoid misleading interpretations of SPA.

**Figure 8 sensors-25-03290-f008:**
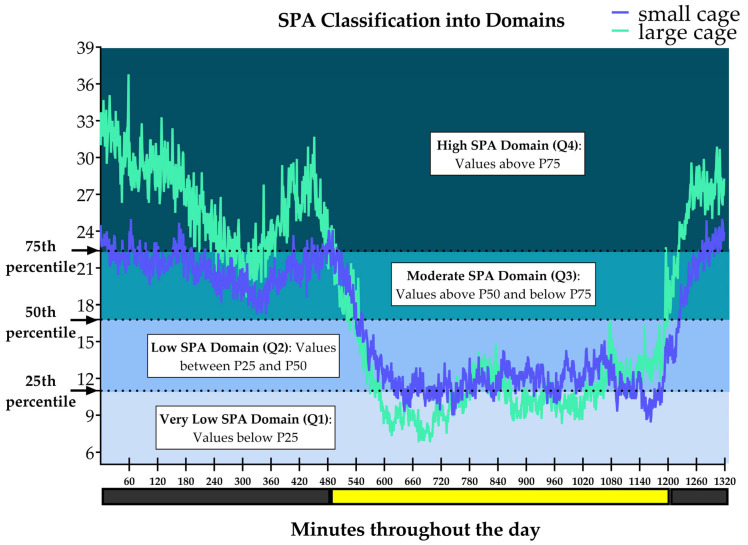
Minute-by-minute spontaneous physical activity (SPA) throughout the data collected across all days in both cages is illustrated, with SPA classified into four domains based on quartile determinations. Values below 11.0 (25th percentile) correspond to the Very Low SPA Domain (VL_SPA_D). Values between 11.0 (25th percentile) and 16.8 (50th percentile) correspond to the Low SPA Domain (L_SPA_D). Values above 16.8 (50th percentile) and below 22.4 (75th percentile) correspond to the Moderate SPA Domain (M_SPA_D). Values above 22.4 (75th percentile), up to the maximum value of 105 (100th percentile), correspond to the High SPA Domain (H_SPA_D).

**Figure 9 sensors-25-03290-f009:**
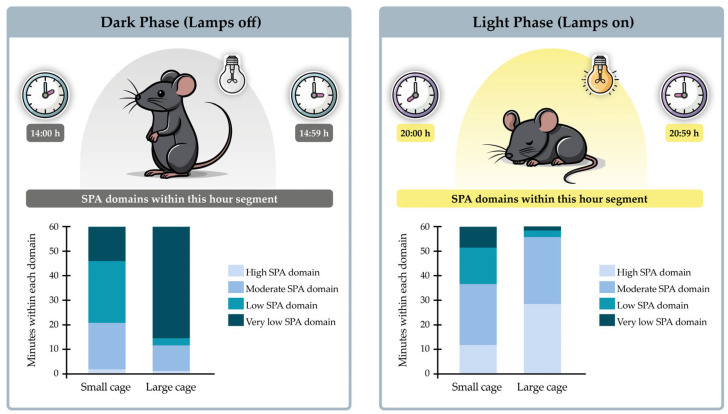
Number of minutes categorized into different activity domains (VL_SPA_D, L_SPA_D, M_SPA_D, and H_SPA_D). The example highlights one hour of the dark phase (14:00 to 14:59), during which the animals are likely awake and active, and one hour of the light phase (20:00 to 20:59), when the animals are likely resting or sleeping. Instead of reporting a single SPA value per hour, our analysis aims to provide details on what happens in regard to SPA within the hours.

**Figure 10 sensors-25-03290-f010:**
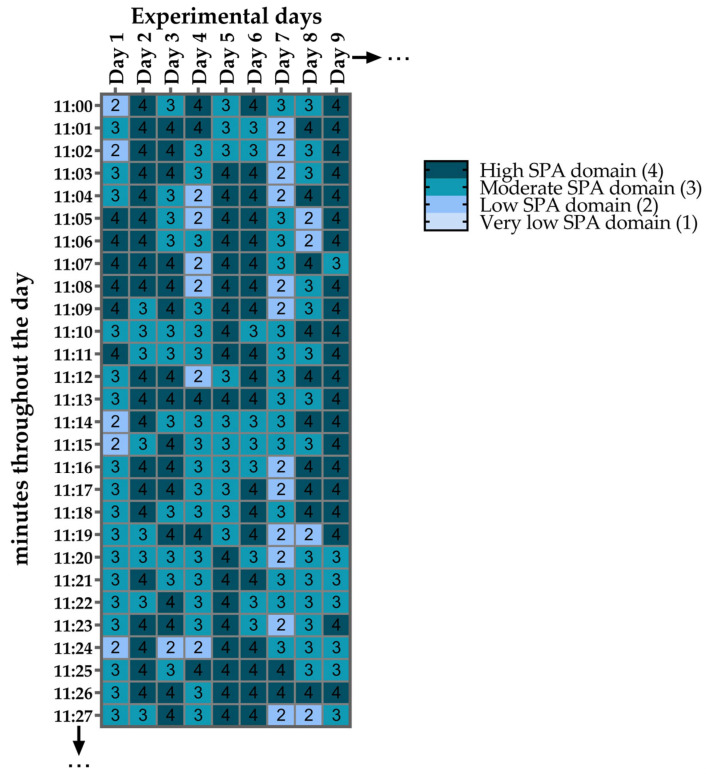
Heat map for visualizing SPA domains minute by minute (y-axis) and across experimental days (x-axis), with each domain assigned a different color.

**Figure 11 sensors-25-03290-f011:**
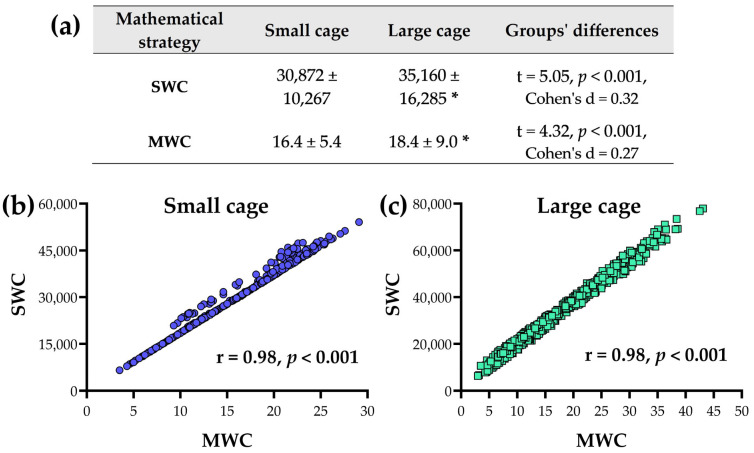
Pairwise comparisons (**a**) between groups (small vs. large cage) for the two mathematical strategies: Sum of Weight Changes (SWC) vs. Mean of Weight Changes (MWC). The relationship between mathematical strategies for the small cage (**b**) and large cage (**c**). The symbol (*) indicates a difference when compared to the small cage. Each group had an n = 506, based on 22 h per day over the full 23-day experiment.

**Figure 12 sensors-25-03290-f012:**
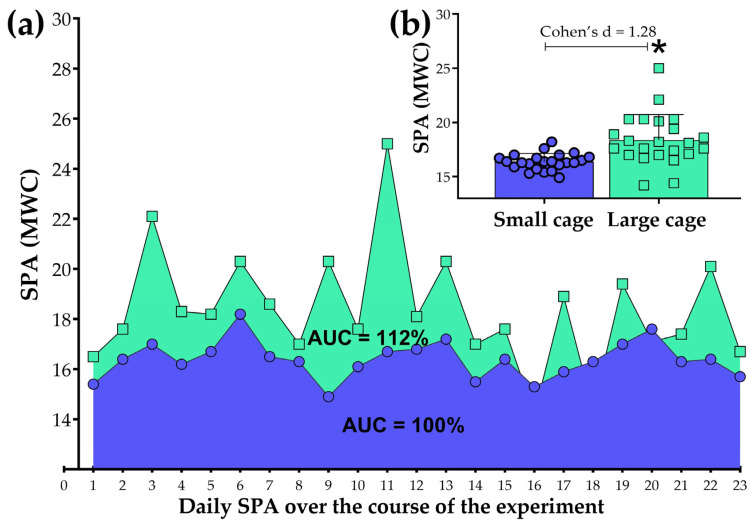
Overall view of daily spontaneous physical activity (SPA) over the course of the present study (**a**). Comparisons between the small cage and large cage groups are presented in the bar graphs (**b**). The accumulated SPA, represented by the area under the curve (AUC), was expressed as a percentage relative to the small cage group (set as 100%) for group comparisons. SPA data for this graph were obtained using the Mean of Weight Changes (MWC) method. The symbol (*) indicates a difference when compared to the small cage group.

**Figure 13 sensors-25-03290-f013:**
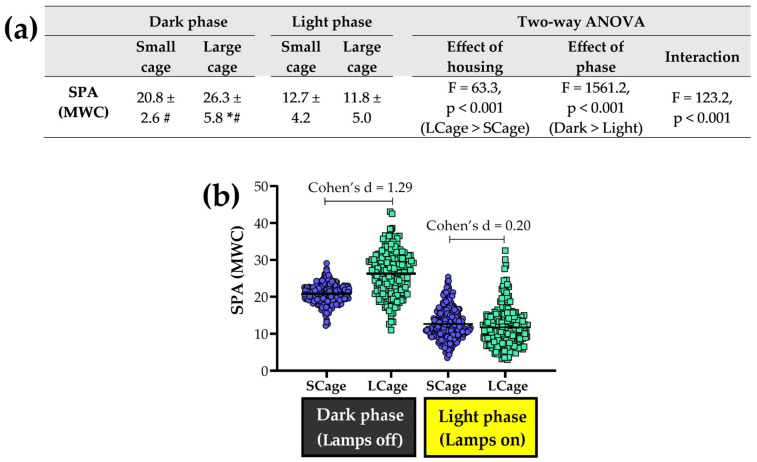
Comparisons of spontaneous physical activity (SPA) and ANOVA results (**a**) between the small cage (SCage) and large cage (LCage) for the two phases: dark vs. light. The symbol (*) indicates a difference when compared to the SCage group within same phase. The symbol (#) indicates a difference when compared to the light phase within same group. Each point on the scatter dot plot represents the SPA value for one hour, with the line indicating the mean (**b**). SPA data for this graph were obtained using the Mean of Weight Changes (MWC) method.

**Figure 14 sensors-25-03290-f014:**
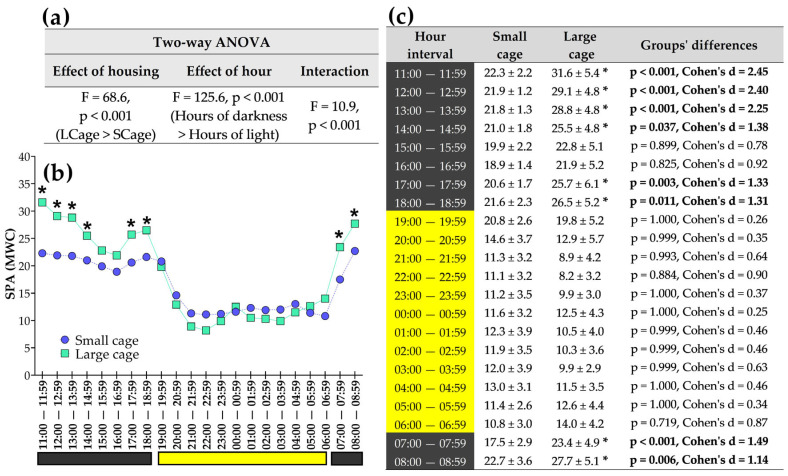
ANOVA comparisons (**a**) between the small cage (SCage) and large cage (LCage) were performed to assess fluctuations in spontaneous physical activity (SPA). Line graphs (**b**) and tables (**c**) were used to represent data for each hourly time interval. The symbol (*) indicates a difference when compared to the small cage within same hour. The black and yellow colors at the bottom of each graph represent the dark and light phases, respectively. SPA data for this graph were obtained using the Mean of Weight Changes (MWC) method.

**Figure 15 sensors-25-03290-f015:**
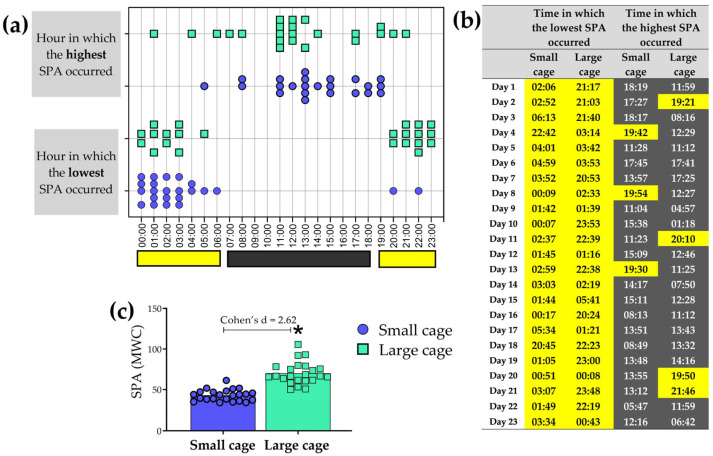
Hours during which the lowest and highest SPA values were observed throughout the day in the small cage (SCage) and large cage (LCage) are shown in (**a**), along with the exact times, including minutes (**b**). Comparisons between groups are presented in the bar graphs, highlighting significant differences in the highest SPA of each day (**c**). The symbol (*) indicates a difference when compared to the small cage. The black and yellow colors at the bottom of each graph represent the dark and light phases, respectively. SPA data for this graph were obtained using the Mean of Weight Changes (MWC) method.

**Figure 16 sensors-25-03290-f016:**
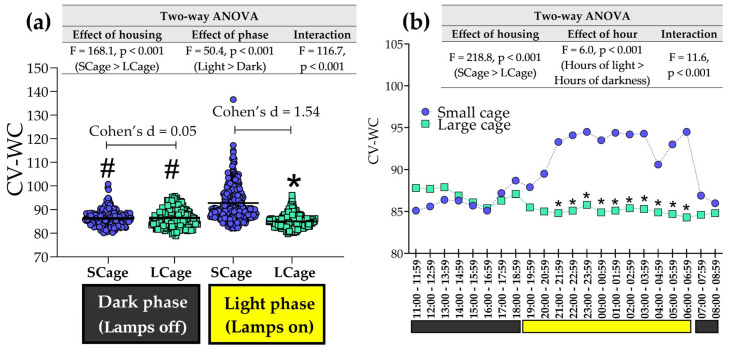
Coefficient of variation of weight changes (CV-WC), organized by phase (dark and light) in (**a**), and by hour (**b**), with comparisons between the small cage (SCage) and large cage (LCage). The symbol (*) indicates a difference when compared to the SCage group within same hour/phase. The symbol (#) indicates a difference when compared to the light phase within same group. The black and yellow colors at the bottom of each graph represent the dark and light phases, respectively.

**Figure 17 sensors-25-03290-f017:**
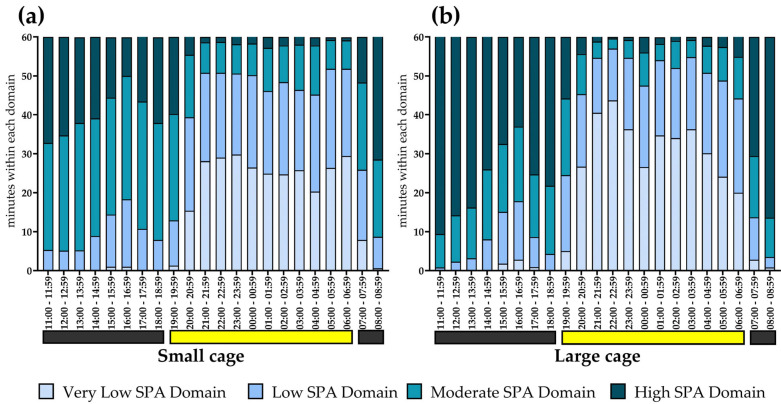
The x-axis represents the time of day in hourly intervals, while the y-axis indicates the total minutes spent within each SPA domain. Panel (**a**) shows the heat map for animals housed in the small cage, while Panel (**b**) displays the heat map for animals housed in the large cage. Light and dark phases are indicated at the bottom of each graph (black for the dark phase and yellow for the light phase). **VL_SPA_D**: Very Low Spontaneous Physical Activity Domain; **L_SPA_D**: Low Spontaneous Physical Activity Domain; **M_SPA_D**: Moderate Spontaneous Physical Activity Domain; and **H_SPA_D**: High Spontaneous Physical Activity Domain. SPA data for this graph were obtained using the Mean of Weight Changes (MWC) method.

**Figure 18 sensors-25-03290-f018:**
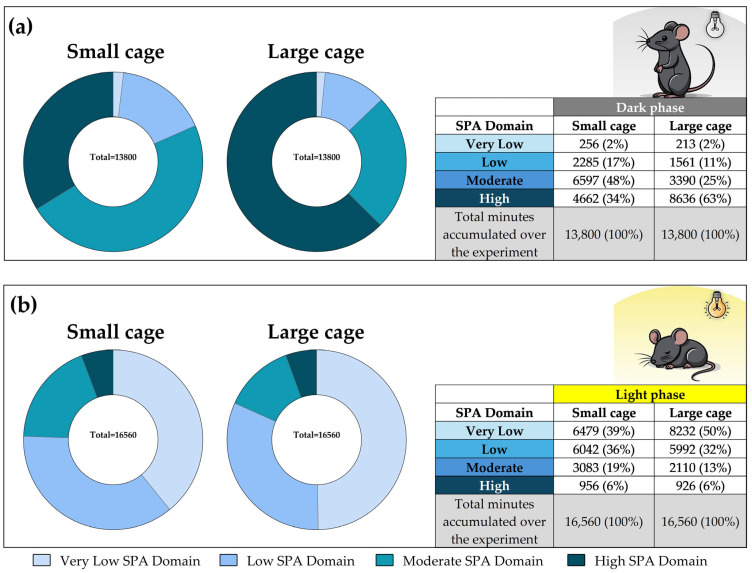
Total accumulated minutes across the four domains during the 23-day experiment. (**a**) shows the dark phase, with a total of 13,800 min; (**b**) shows the light phase, with a total of 16,560 min. The pie charts summarize the distribution of time spent in each of the four domains. SPA data for this graph were obtained using the Mean of Weight Changes (MWC) method.

**Figure 19 sensors-25-03290-f019:**
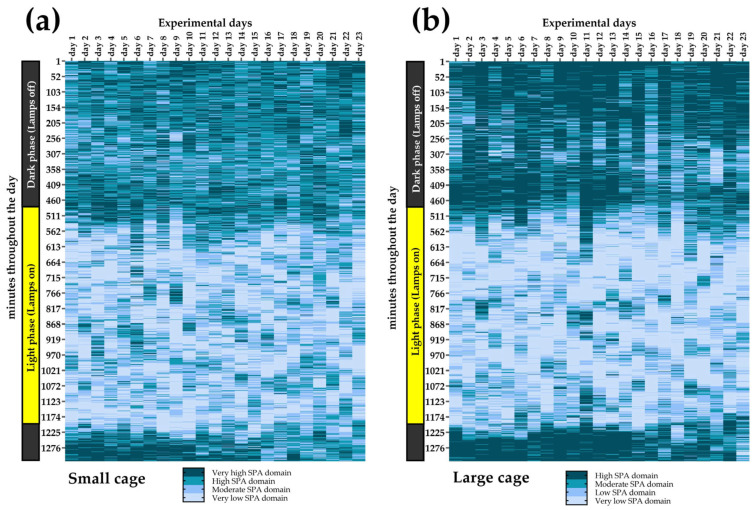
Minute-by-minute heat maps of SPA domains intraday (y-axes) and over days (x axes) for rodents housed in the small (**a**) and large cages (**b**). The dark phase (lights off) and light phase (lights on) are highlighted on the left for reference. SPA data for this graph were obtained using the Mean of Weight Changes (MWC) method.

**Table 1 sensors-25-03290-t001:** Standard deviations of weight changes (SD-WC) for small and large cage conditions across different phases (dark, light, and overall).

	Dark Phase		Light Phase		All Phases	
	SCage	LCage	SCage	LCage	SCage	LCage
**SD-WC**	18.1 ± 2.6	23.1 ± 5.7	11.3 ± 3.6	10.1 ± 4.5	14.4 ± 4.6	16.0 ± 8.3

**Table 2 sensors-25-03290-t002:** Minutes animals spent in each SPA domain during each hourly interval. The sum of the minutes spent in each domain per hour will always equal 60 min.

	Small Cage				Large Cage			
	VL_SPA_D	L_SPA_D	M_SPA_D	H_SPA_D	VL_SPA_D	L_SPA_D	M_SPA_D	H_SPA_D
** 11:00–11:59 **	0.1 ± 0.6	5.2 ± 6.0	27.5 ± 8.4 &¥	27.2 ± 13.4 &¥	0.0 ± 0.0	0.8 ± 1.2	8.6 ± 7.5 &¥$	50.7 ± 8.3 &¥£
** 12:00–12:59 **	0.0 ± 0.0	5.1 ± 4.7 &	29.6 ± 6.4 &¥	25.3 ± 8.4 &¥	0.0 ± 0.0	2.3 ± 4.5	11.9 ± 9.9 &¥$	45.9 ± 13.3 &¥£
** 13:00–13:59 **	0.0 ± 0.0	5.2 ± 4.5 &	32.7 ± 6.2 &¥	22.0 ± 7.5 &¥	0.0 ± 0.0	3.2 ± 4.4	13.0 ± 9.4 &¥$	43.8 ± 12.8 &¥£
** 14:00–14:59 **	0.1 ± 0.5	8.8 ± 8.0 &	30.2 ± 6.8 &¥	20.9 ± 10.4 &¥	0.2 ± 0.5	7.8 ± 8.8 &	18.0 ± 8.9 &¥	34.1 ± 14.6 &¥
** 15:00–15:59 **	1.0 ± 2.3	13.4 ± 7.7 &	30.0 ± 7.2 &¥	15.5 ± 10.2 &£	1.8 ± 2.9	13.3 ± 11.1 &	17.4 ± 7.2 &	27.5 ± 17.0 &
** 16:00–16:59 **	1.0 ± 0.9	17.3 ± 6.0 &	31.7 ± 6.0 &	9.9 ± 7.1 &£	2.8 ± 7.2	15.0 ± 10.6 &	19.2 ± 8.5 &	23.0 ± 16.7 &
** 17:00–17:59 **	0.2 ± 0.7	10.5 ± 6.2 &	32.7 ± 6.2 &¥	16.6 ± 8.1 &£	0.9 ± 3.2	7.7 ± 11.3 &	16.1 ± 8.9 &$	35.3 ± 17.3 &¥£
** 18:00–18:59 **	0.2 ± 0.9	7.7 ± 7.1 &	30.0 ± 6.5 &¥	22.0 ± 10.3 &¥	0.0 ± 0.0	4.3 ± 6.1	17.5 ± 10.6 &	38.2 ± 15.1 &¥£
19:00–19:59	1.3 ± 3.2	11.6 ± 8.9 &	27.3 ± 5.9 &¥	19.8 ± 10.8 &	5.0 ± 7.4	19.5 ± 13.1 &	19.7 ± 9.6 &	15.8 ± 14.1 &
20:00–20:59	15.4 ± 15.0	24.0 ± 9.4	16.0 ± 10.1	4.7 ± 6.1 &¥£	26.7 ± 19.8	18.6 ± 9.7	10.3 ± 10.4	4.4 ± 8.4 &¥
21:00–21:59	28.1 ± 16.0	22.7 ± 9.4	7.8 ± 7.1 &¥	1.4 ± 2.4 &¥	40.5 ± 18.4	14.1 ± 10.7 &	4.2 ± 7.2 &¥	1.3 ± 3.1 &¥
22:00–22:59	29.0 ± 14.5	21.8 ± 8.4	7.9 ± 7.8 &¥	1.3 ± 3.3 &¥£	43.7 ± 15.4	13.3 ± 11.8 &	2.6 ± 4.5 &¥	0.4 ± 0.9 &¥
23:00–23:59	29.8 ± 14.5	20.8 ± 8.1	7.5 ± 7.7 &¥	2.0 ± 4.6 &¥	36.3 ± 15.4	18.3 ± 11.5 &	4.6 ± 6.1 &¥	0.8 ± 1.3 &¥
00:00–00:59	26.5 ± 15.9	23.7 ± 10.5	8.1 ± 7.0 &¥	1.8 ± 3.1 &¥	26.6 ± 15.8	20.9 ± 10.3	8.5 ± 7.2 &¥	4.0 ± 6.6 &¥
01:00–01:59	24.9 ± 16.0	21.2 ± 8.2	11.1 ± 9.7	2.8 ± 3.9 &¥£	34.7 ± 15.8	19.3 ± 11.8	4.2 ± 4.7 &¥	1.7 ± 5.0 &¥
02:00–02:59	24.7 ± 15.2	23.7 ± 9.5	9.4 ± 8.3 &¥	2.2 ± 4.5 &¥£	34.0 ± 17.4	18.0 ± 10.9	7.0 ± 7.8 &¥	1.1 ± 1.9 &¥
03:00–03:59	25.8 ± 17.6	20.6 ± 8.4	11.6 ± 10.7	2.0 ± 2.6 &¥£	36.3 ± 14.8	18.5 ± 10.7	4.4 ± 4.9 &¥	0.8 ± 1.4 &¥
04:00–04:59	20.3 ± 13.3	24.9 ± 7.8	12.6 ± 9.9	2.1 ± 2.3 &¥£	30.1 ± 14.7	20.7 ± 9.0	6.9 ± 6.2 &¥	2.3 ± 2.8 &¥
05:00–05:59	26.4 ± 12.9	25.4 ± 9.4	7.4 ± 5.6 &¥	0.7 ± 1.2 &¥£	24.1 ± 16.2	24.7 ± 10.4	8.6 ± 7.3 &¥	2.7 ± 7.5 &¥
06:00–06:59	29.4 ± 14.3	22.4 ± 9.1	7.3 ± 7.4 &¥	0.8 ± 1.4 &¥£	20.0 ± 15.5	24.2 ± 11.0	10.7 ± 7.0 ¥	5.1 ± 7.5 &¥
** 07:00–07:59 **	7.9 ± 9.4	18.0 ± 7.7 &	22.4 ± 8.1 &	11.7 ± 8.0	2.8 ± 4.5	10.9 ± 8.7	15.7 ± 8.2 &	30.6 ± 15.1 &¥$
** 08:00–08:59 **	0.6 ± 1.4	8.1 ± 9.6 &	19.8 ± 11.0 &¥	31.6 ± 18.5 &¥	0.8 ± 4.0	2.7 ± 6.1	10.1 ± 8.5 &¥	46.4 ± 15.2 &¥£

**VL_SPA_D**: Very Low Spontaneous Physical Activity Domain; **L_SPA_D**: Low Spontaneous Physical Activity Domain; **M_SPA_D**: Moderate Spontaneous Physical Activity Domain; and **H_SPA_D**: High Spontaneous Physical Activity Domain. A rank transformation was applied to the results presented in this table; therefore, the findings should be interpreted as exploratory. The symbol (&) indicates a difference when compared to the VL_SPA_D within same group. The symbol (¥) indicates a difference when compared to the L_SPA_D within same group. The symbol (£) indicates a difference when compared to the M_SPA_D within same group. The symbol ($) indicates a difference when compared to the small cage within the same domain.

**Table 3 sensors-25-03290-t003:** The number of minutes spent in each SPA domain across all experimental days, considering only the dark phase. The total duration of the dark phase (600 min) was set as 100%, and the percentage contribution of each domain was calculated.

	Dark Phase							
	Small Cage				Large Cage			
Day	VLSPAD	LSPAD	MSPAD	HSPAD	VLSPAD	LSPAD	MSPAD	HSPAD
**1st**	7 (1%)	156 (26%)	283 (47%)	154 (26%)	7 (1%)	100 (17%)	172 (29%)	321 (54%)
**2nd**	2 (0%)	107 (18%)	269 (45%)	222 (37%)	5 (1%)	64 (11%)	141 (24%)	390 (65%)
**3rd**	3 (1%)	79 (13%)	234 (39%)	284 (47%)	10 (2%)	35 (6%)	62 (10%)	493 (82%)
**4th**	7 (1%)	113 (19%)	282 (47%)	198 (33%)	18 (3%)	59 (10%)	163 (27%)	360 (60%)
**5th**	4 (1%)	99 (17%)	298 (50%)	199 (33%)	7 (1%)	88 (15%)	128 (21%)	377 (63%)
**6th**	11 (2%)	78 (13%)	242 (40%)	269 (45%)	2 (0.3%)	32 (5%)	103 (17%)	463 (77%)
**7th**	7 (1%)	85 (14%)	276 (46%)	232 (39%)	5 (1%)	31 (5%)	112 (19%)	452 (75%)
**8th**	27 (5%)	105 (18%)	219 (37%)	249 (42%)	0 (0%)	54 (9%)	187 (31%)	359 (60%)
**9th**	36 (6%)	98 (16%)	258 (43%)	208 (35%)	3 (1%)	48 (8%)	106 (18%)	443 (74%)
**10th**	27 (5%)	76 (13%)	243 (41%)	254 (42%)	2 (0.3%)	64 (11%)	173 (29%)	361 (60%)
**11th**	1 (0.2%)	118 (20%)	303 (51%)	178 (30%)	0 (0%)	4 (1%)	56 (9%)	540 (90%)
**12th**	20 (3%)	69 (12%)	269 (45%)	242 (40%)	7 (1%)	65 (11%)	118 (20%)	410 (68%)
**13th**	0 (0%)	67 (11%)	307 (51%)	226 (38%)	0 (0%)	12 (2%)	65 (11%)	523 (87%)
**14th**	8 (1%)	132 (22%)	331 (55%)	129 (22%)	2 (0.3%)	86 (14%)	208 (35%)	304 (51%)
**15th**	32 (5%)	65 (11%)	296 (49%)	207 (35%)	12 (2%)	56 (9%)	123 (21%)	409 (68%)
**16th**	9 (2%)	107 (18%)	341 (57%)	143 (24%)	24 (4%)	182 (30%)	234 (39%)	160 (27%)
**17th**	4 (1%)	117 (20%)	297 (50%)	182 (30%)	0 (0%)	50 (8%)	162 (27%)	388 (65%)
**18th**	2 (0.3%)	89 (15%)	322 (54%)	187 (31%)	24 (4%)	168 (28%)	211 (35%)	197 (33%)
**19th**	6 (1%)	127 (21%)	301 (50%)	166 (28%)	4 (1%)	57 (10%)	185 (31%)	354 (59%)
**20th**	7 (1%)	84 (14%)	308 (51%)	201 (34%)	7 (1%)	66 (11%)	195 (33%)	332 (55%)
**21th**	8 (1%)	102 (17%)	329 (55%)	161 (27%)	50 (8%)	96 (16%)	165 (28%)	289 (48%)
**22th**	24 (4%)	81 (14%)	268 (45%)	227 (38%)	0 (0%)	29 (5%)	139 (23%)	432 (72%)
**23th**	4 (1%)	131 (22%)	321 (54%)	144 (24%)	24 (4%)	115 (19%)	182 (30%)	279 (47%)
**Overall** **average**	**11 ± 11**	**99 ± 24 &**	**287 ± 33 &**¥	**203 ± 42 &**¥£	**9 ± 12**	**68 ± 43 &**	**147 ± 49 *&**¥	**375 ± 94 *&**¥£

The symbol (*****) indicates a difference when compared to the small cage group within same domain. The symbol (&) indicates a difference when compared to the VL_SPA_D within same group. The symbol (¥) indicates a difference when compared to the L_SPA_D within same group. The symbol (£) indicates a difference when compared to the M_SPA_D within same group. **VL_SPA_D**: Very Low Spontaneous Physical Activity Domain; **L_SPA_D**: Low Spontaneous Physical Activity Domain; **M_SPA_D**: Moderate Spontaneous Physical Activity Domain; and **H_SPA_D**: High Spontaneous Physical Activity Domain. SPA data for this graph were obtained using the Mean of Weight Changes (MWC) method.

**Table 4 sensors-25-03290-t004:** The number of minutes spent in each SPA domain across all experimental days, considering only the light phase. The total duration of the light phase (720 min) was set as 100%, and the percentage contribution of each domain was calculated.

	Light Phase							
	Small Cage				Large Cage			
Day	VL_SPA_D	L_SPA_D	M_SPA_D	H_SPA_D	VL_SPA_D	L_SPA_D	M_SPA_D	H_SPA_D
**1st**	359 (50%)	256 (36%)	82 (11%)	23 (3%)	404 (56%)	223 (31%)	62 (9%)	31 (4%)
**2nd**	340 (47%)	223 (31%)	111 (15%)	46 (6%)	378 (53%)	242 (34%)	80 (11%)	20 (3%)
**3rd**	343 (48%)	216 (30%)	96 (13%)	65 (9%)	298 (41%)	211 (29%)	136 (19%)	75 (10%)
**4th**	317 (44%)	255 (35%)	103 (14%)	45 (6%)	308 (43%)	262 (36%)	126 (18%)	24 (3%)
**5th**	237 (33%)	310 (43%)	136 (19%)	37 (5%)	409 (57%)	224 (31%)	60 (8%)	27 (4%)
**6th**	224 (31%)	204 (28%)	176 (24%)	116 (16%)	291 (40%)	259 (36%)	93 (13%)	77 (11%)
**7th**	308 (43%)	248 (34%)	135 (19%)	29 (4%)	436 (61%)	204 (28%)	55 (8%)	25 (3%)
**8th**	325 (45%)	253 (35%)	99 (14%)	43 (6%)	405 (56%)	233 (32%)	64 (9%)	18 (3%)
**9th**	399 (55%)	202 (28%)	87 (12%)	32 (4%)	308 (43%)	295 (41%)	98 (14%)	19 (3%)
**10th**	330 (46%)	224 (31%)	127 (18%)	39 (5%)	330 (46%)	278 (39%)	93 (13%)	19 (3%)
**11th**	240 (33%)	288 (40%)	138 (19%)	54 (8%)	241 (33%)	144 (20%)	126 (18%)	209 (29%)
**12th**	237 (33%)	285 (40%)	156 (22%)	42 (6%)	430 (60%)	178 (25%)	86 (12%)	26 (4%)
**13th**	233 (32%)	283 (39%)	145 (20%)	59 (8%)	430 (60%)	174 (24%)	90 (13%)	26 (4%)
**14th**	306 (43%)	225 (31%)	145 (20%)	44 (6%)	352 (49%)	244 (34%)	101 (14%)	23 (3%)
**15th**	243 (34%)	320 (44%)	128 (18%)	29 (4%)	432 (60%)	218 (30%)	52 (7%)	18 (3%)
**16th**	334 (46%)	231 (32%)	127 (18%)	28 (4%)	424 (59%)	246 (34%)	44 (6%)	6 (1%)
**17th**	323 (45%)	244 (34%)	111 (15%)	42 (6%)	347 (48%)	238 (33%)	97 (13%)	38 (5%)
**18th**	238 (33%)	319 (44%)	145 (20%)	18 (3%)	477 (66%)	209 (29%)	23 (3%)	11 (2%)
**19th**	170 (24%)	350 (49%)	166 (23%)	34 (5%)	253 (35%)	248 (34%)	172 (24%)	47 (7%)
**20th**	156 (22%)	334 (46%)	183 (25%)	47 (7%)	374 (52%)	178 (25%)	115 (16%)	53 (7%)
**21th**	252 (35%)	244 (34%)	179 (25%)	45 (6%)	234 (33%)	320 (44%)	131 (18%)	35 (5%)
**22th**	251 (35%)	276 (38%)	174 (24%)	19 (3%)	303 (42%)	234 (33%)	117 (16%)	66 (9%)
**23th**	314 (44%)	252 (35%)	134 (19%)	20 (3%)	368 (51%)	230 (32%)	89 (12%)	33 (5%)
**Overall average**	**282 ± 62**	**263 ± 42**	**134 ± 30 &**¥	**42 ± 20 &**¥£	**358 ± 69 ***	**230 ± 40 &**	**92 ± 35 *&**¥	**40 ± 41 &**¥£

The symbol (*****) indicates a difference when compared to the small cage group within same domain. The symbol (&) indicates a difference when compared to the VL_SPA_D within same group. The symbol (¥) indicates a difference when compared to the L_SPA_D within same group. The symbol (£) indicates a difference when compared to the M_SPA_D within same group. **VL_SPA_D**: Very Low Spontaneous Physical Activity Domain; **L_SPA_D**: Low Spontaneous Physical Activity Domain; **M_SPA_D**: Moderate Spontaneous Physical Activity Domain; and **H_SPA_D**: High Spontaneous Physical Activity Domain. SPA data for this graph were obtained using the Mean of Weight Changes (MWC) method.

## Data Availability

The raw data supporting the conclusions of this article will be made available by the authors on request.

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
