# Peer review of "Exploring New Ways to Analyze Data on the Spontaneous Physical Activity of Rodents Through a Weighing Balance"

_sensors, 2025, doi:10.3390/s25113290_

Round 1
Reviewer 1 Report
Comments and Suggestions for Authors
This study proposed and constructed a weighing balance system for measuring weight changes of rodents in a cage, and also used new analytical strategies, such as using the mean of weight changes (MWC), assessing the dispersion of weight changes, and classifying SPA into domains to enhance data interpretation.
The manuscript is valuable to be published in the journal for two reasons. (1) The weighing balance system was well described enough to enable others to build their own systems. (2) And the system and analytical measures (MWC and classification of SPA into four domains) were verified by an experiment using small cage and large cage during dark and light conditions.
However, for clear understanding of the readers and more scientific analysis, it is suggested to consider the following opinion.
(1) p.15, line 365-368 : The study used an independent t-test to compare the SCage and LCage and also conducted the two-way ANOVA test on SPA data to determine the effects of housing conditions (SCage vs. LCage) and phase (dark vs. light). For my understanding, the independent t-test was not necessary if you conduct two-way ANOVA because the two-way ANOVA can test the main effect of the housing condition (SCage vs. LCage).
(2) For future study, I suggest the 'center of mass' data to be used along with weight based measures to measure SPA. The center of mass data have been used very often in the human walking or standing studies by using force plate system. I think the balance system of the manuscript is very similar to the force plate system for human study, and the center of mass (position) can be calculated by the data collected from the balance system.
Author Response
Comments 1: This study proposed and constructed a weighing balance system for measuring weight changes of rodents in a cage, and also used new analytical strategies, such as using the mean of weight changes (MWC), assessing the dispersion of weight changes, and classifying SPA into domains to enhance data interpretation. The manuscript is valuable to be published in the journal for two reasons. (1) The weighing balance system was well described enough to enable others to build their own systems. (2) And the system and analytical measures (MWC and classification of SPA into four domains) were verified by an experiment using small cage and large cage during dark and light conditions. |
Response 1: We sincerely thank the reviewer for highlighting the strengths of our manuscript. We are pleased that the weighing balance system was found to be clearly described.
|
Comments 2: p.15, line 365-368 : The study used an independent t-test to compare the SCage and LCage and also conducted the two-way ANOVA test on SPA data to determine the effects of housing conditions (SCage vs. LCage) and phase (dark vs. light). For my understanding, the independent t-test was not necessary if you conduct two-way ANOVA because the two-way ANOVA can test the main effect of the housing condition (SCage vs. LCage). |
Response 2: We sincerely thank the reviewer for the comment. We agree that the findings from the independent t-test support the same conclusion as the main effect of housing observed in the ANOVA results presented in Figures 13 and 14. Both statistical tests converge in showing that animals housed in the LCage were more physically active than those housed in the SCage. Although may seem redundant, we chose to include both statistical approaches to illustrate different analytical options for researchers interested in working with SPA, highlighting that the choice of statistical test should be guided by the characteristics of the dataset. By using the independent t-test, we aimed to demonstrate that, despite its simplicity, this test can be highly valuable. The t-test is the most appropriate option for certain types of data, particularly those with lower temporal resolution (e.g., a single daily value, as illustrated in Figure 12) or in contexts where the main goal is purely to compare activity between experimental groups, without concerns about when the activity occurs (as in Figure 11). On the other hand, when we used ANOVA, our goal was to show that this test is more appropriate when researchers work with data categorized by temporal factors, such as light and dark phases or hour-by-hour fluctuations. We aimed to gradually guide the reader step by step from a simple comparison (t-test) to a complex analysis (ANOVA) that captures temporal patterns. We have clarified this rationale in lines 365 to 371.
|
Comments 3: For future study, I suggest the 'center of mass' data to be used along with weight based measures to measure SPA. The center of mass data have been used very often in the human walking or standing studies by using force plate system. I think the balance system of the manuscript is very similar to the force plate system for human study, and the center of mass (position) can be calculated by the data collected from the balance system. Response 3: We fully agree with the reviewer’s insightful suggestion. We acknowledge that it is indeed feasible to estimate the animals’ position by analyzing the signals from the three load cells, particularly when a calibration system is applied across the different quadrants of the platform. To address this valuable suggestion from the reviewer, we have incorporated the following sentence into the Discussion section: “An important point to highlight is that weight-based SPA assessments still hold significant untapped potential. For example, future studies could estimate the animal’s position (center of mass) using load cell signals, thereby enriching SPA analysis with spatial information and enabling more refined assessments of movement trajectories and exploratory patterns within the cage.”
|

Reviewer 2 Report
Comments and Suggestions for Authors
Menezes Scariot et al have conducted an insightful study on measuring rodents' spontaneous physical activity. They introduced a method that tracks changes in the weight balance of the cage caused by the rodents' movements. The study is well-written, with the methods clearly and comprehensively explained. However, the length of the study may be a bit extensive for some readers.
Here are some comments:
Page 8, line 217: How many days are the “several days”?
Page 11, Equations 2 and 3: The SPA abbreviation should be removed from the equations because it is typically not used with SWC and MWC later in the manuscript.
Page 13, Equation 4: The ‘SPA’ should likely be the ‘MWC’. What where the cut-points for the SWC?
Page 16, Figure 11. Why the points are not in the straight line, due to variations in the sampling frequency?
Page 16, line 397. The area under the curve needs some other term abbreviation than ‘AUC’. It reminds too much area under the curve (AUC) used in the receiver operating characteristic analysis (ROC). Maybe it could be just integral?
Page 17, Figure 12. The figure 12 c) is not necessarily, the result is already shown in the Figure 12 b). The 12 % higher result can be just said in text.
Page 22, Table 2. The table values can be shown in the supplemental material as the results are shown in the Figure 17. The statistical comparisons can done with Kruskal-Wallis test.
Discussion: Is the proposed method scalable to other number of rodents, or to other species? How sensitive is the MWC value to sampling rate?
Author Response
Comments 1: Menezes Scariot et al have conducted an insightful study on measuring rodents' spontaneous physical activity. They introduced a method that tracks changes in the weight balance of the cage caused by the rodents' movements. The study is well-written, with the methods clearly and comprehensively explained. However, the length of the study may be a bit extensive for some readers. |
Response 1: We sincerely thank the reviewer for the generous and encouraging assessment of our work. We are pleased to know that the clarity and depth of our methodological description were appreciated. Regarding the manuscript's length, we acknowledge that the text may appear extensive; however, this was a deliberate choice. Given the technical nature of the method and the multiple steps involved in detecting and presenting SPA, we aimed to construct a didactic and well-structured manuscript that leads readers to a comprehensive understanding in a progressive manner. The inclusion of numerous tables and figures was intentional, aiming to offer readers a clear, complete, and transparent view of the data. We believe this level of detail not only enhances understanding but also highlights the richness of the dataset. Ultimately, our goal was to produce a manuscript that serves as a guide for researchers aiming to understand and replicate the procedures involved in using weighing-based systems for movement detection.
|
Comments 2: Page 8, line 217: How many days are the “several days”? |
Response 2: We appreciate the reviewer’s comment. "Several days" refers to a period of one-month. We will clarify this in the manuscript to avoid any ambiguity. Thank you for pointing that out.
|
Comments 3: Page 11, Equations 2 and 3: The SPA abbreviation should be removed from the equations because it is typically not used with SWC and MWC later in the manuscript. Response 3: We thank the reviewer for this helpful observation. As suggested, we have removed the SPA abbreviation from Equations 2 and 3.
Comments 4: Page 13, Equation 4: The ‘SPA’ should likely be the ‘MWC’. What where the cut-points for the SWC? Response 4: Thank you for this valuable feedback. The reviewer is correct in noting that the domain analyses were conducted using MWC. The values of 11.0, 16.8, and 22.4, which correspond to the 25th percentile (P25), 50th percentile (median, P50), and 75th percentile (P75), respectively, were indeed derived from MWC (not from SWC). This distinction has been more prominently emphasized in the revised manuscript to ensure greater clarity. We did not conduct domain analyses using SWC, since the associations confirmed that MWC sufficiently serves as a valid measure.
Comments 5: Page 16, Figure 11. Why the points are not in the straight line, due to variations in the sampling frequency? Response 5: We sincerely thank the reviewer for their excellent attention to detail in noticing this. We believe the reviewer is referring to Figures 11b and 11c, where the points do not align perfectly along the correlation line. In agreement with the reviewer, we believe that the small deviation from a perfect correlation is likely due to variations in the sampling frequency. Since the SWC method does not account for the number of data points per minute, this can result in slightly overestimated SPA when more data points are obtained. However, this is not necessarily problematic, since the correlation coefficient of 0.98 (p<0.001) demonstrates a strong and meaningful association between the mathematical strategies. In fact, the development of the MWC strategy was specifically aimed at addressing these sampling rate inconsistencies, which are inherent in simpler signal acquisition systems such as the ARDUINO system.
Comments 6: Page 16, line 397. The area under the curve needs some other term abbreviation than ‘AUC’. It reminds too much area under the curve (AUC) used in the receiver operating characteristic analysis (ROC). Maybe it could be just integral? Response 6: We thank the reviewer for this valuable comment. We understand the concern regarding the use of AUC (area under the curve). The suggestion to use the term integral is appreciated; however, we believe that it does not fully reflect the specific nature of our analysis. A definite integral refers to the exact calculation of the area under the curve of a continuous mathematical function, such as f(x)=sin(x), for example. However, in experimental contexts, however, where the function is unknown or cannot be explicitly determined (as with our SPA data), numerical methods like the trapezoidal rule are employed to approximate this area. Although the trapezoidal method approximates the result of an integral and offers a practical solution for estimating the area under a curve, it does not constitute a formal integral in the strict mathematical sense. We hope this explanation clarifies why we have chosen to retain the term AUC in the manuscript. Moreover, the term AUC is more accurate and widely accepted in various fields—such as pharmacokinetics and nutrition research to quantify cumulative values over time [1].
Comments 7: Page 17, Figure 12. The figure 12 c) is not necessarily, the result is already shown in the Figure 12 b). The 12 % higher result can be just said in text. Response 7: We appreciate the reviewer’s observation and agree that panel 12c is redundant, as its message is already visually conveyed by panel 12b. Therefore, we have removed panel 12c and incorporated the information about the AUC directly into the colored area of panel 12a.
Comments 8: Page 22, Table 2. The table values can be shown in the supplemental material as the results are shown in the Figure 17. The statistical comparisons can done with Kruskal-Wallis test. Response 8: Initially, we considered using the Kruskal–Wallis test. However, this test is inherently one-way and does not allow for the evaluation of two main effects in our factorial design [2]. Using it would have required us to disregard one factor or collapse across it, which would violate the structure of our question proposed in table 2. Inspired by the reviewer's excellent question, we revisited the analysis and explored alternative procedures, and found the rank transformation to be the most accessible [3]. Although not a purely nonparametric test, this approach is useful for data that violate parametric assumptions. To implement it, the entire set of observations is ranked from smallest to largest—assigning rank 1 to the smallest value, rank 2 to the next smallest, and so on up to the maximum value, which receives the highest rank. The ranked data can then be analyzed using standard parametric methods (for example, a two‑way factorial ANOVA). This enables us to evaluate both the effect of housing and the effect of domain simultaneously. Given these considerations, we chose to retain Table 2—now as a statistical summary—allowing readers to visualize and compare the time (minutes per hour) that animals spent in each domain throughout the day.
Comments 9: Discussion: Is the proposed method scalable to other number of rodents, or to other species? How sensitive is the MWC value to sampling rate? Response 9: We thank the reviewer for raising this important point. The scalability of the proposed method to different numbers of rodents or other species primarily depends on the full capacity of the three load cells used in our system. In our case, the total capacity is 15 kg, which represents the safe load limit. After subtracting the weight of the cage, platform (UMP), bedding, food, and water, the system could still support approximately 4.5 kg of animal weight. This would be sufficient to accommodate up to 15 adult Wistar rats, each weighing approximately 300 g. In the case of smaller species such as mice, which typically weigh ~30 g, the system could theoretically support up to 150 animals. However, these estimates are strictly based on the load cell’s maximum capacity, and space availability within the cage and ethical guidelines concerning animal density and welfare must also be considered. Thus, the balance we designed is versatile and scalable for various experimental contexts involving rodents. However, for heavier species, the same structural design should be adapted by using load cells with a higher capacity (e.g., 50 or 100 kg). While these load cells can support more weight, it's important to note that they may lose sensitivity to more subtle movements. We appreciate the reviewer’s interest in the sensitivity of the MWC (Mean Weight Change) to the sampling rate. The MWC is less sensitive to sampling rate variations. This is because the MWC is calculated by dividing the sum of the weight changes by the number of samples, effectively correcting for discrepancies in the sampling frequency. The following hypothetical example demonstrates that, even with a reduction in the sampling rate, the MWC remains virtually unchanged due to its normalization by the number of samples.
Scenario 1 (Regular Sampling Rate): · Sampling frequency: 10 Hz (10 samples per second). In one minute, we have 600 samples. · Let’s assume a similar movement pattern, where the variation between consecutive samples remains constant at 10 units. · Since the variation is 10 units, the sum of the weight changes is 5990. In this scenario, dividing this sum by the number of samples (600) results in an MWC of 9.9.
Scenario 2 (Irregular Sampling Rate): · Sampling frequency: 5 Hz (5 samples per second). Now, we have 300 samples in one minute. · The variation between consecutive samples remains the same (10 units). · In this scenario, when we divide the sum of the weight changes (2990) by the number of samples (300), the MWC still comes out to 9.9.
References
|

Reviewer 3 Report
Comments and Suggestions for Authors
The article presents the following concerns:
- Move highlights to the introduction.
- Mention the main quantitative results in the abstract.
- Add hyperlinks to references, tables, and figures.
- At the end of the introduction, add a paragraph describing the manuscript's structure.
- Add a brief description between section and subsection titles.
- In Figures 1 and 2, decimal division is indicated by a comma; please correct it.
- The resolution of the figures needs to be improved.
- Please define all variables and parameters in the equations and figures in the main text.
- The article does not include any calculations or reasoning to justify why 10 animals per group (n = 20) are sufficient for the statistical analyses performed.
- "Newly weaned mice" are mentioned, but the inclusion criteria are not clearly defined, nor is it reported whether animals were discarded or for what reasons. A better description of the selected animals is needed.
- The density per animal in the cages varies between groups. This introduces an additional variable that is not controlled separately from cage size.
- The proposed system is not compared to another reference method, which weakens claims about its validity.
- The system is inferred to be "robust and valid." However, validation is limited to detecting expected differences between large and small cages or between light/dark phases, representing a circular analysis.
- No confidence intervals or effect sizes are reported (except once with Cohen's d). This prevents assessment of the true magnitude of the differences.
- There is no evidence of verification of compliance with the assumptions of homogeneity of variances or discussion of possible additional transformations beyond the inverse normalization.
- Table 2 lacks statistical analysis due to non-normality. Nonparametric methods could have been used for at least an exploratory comparison between hours.
- How signals are synchronized or how precise temporal alignment between cells is ensured is not specified. The conversion of raw signal to weight units is explained, but the calibration standard errors are not detailed.
- Variability in sampling frequency (unstable 40 Hz). If not properly controlled, this can introduce artifacts. MWC mitigates this, but its impact on longitudinal studies is not analyzed.
Author Response
Comments 1: Dear authors, Move highlights to the introduction. Response 1: Thank you for your comment. However, we followed the journal's submission guidelines (https://www.mdpi.com/journal/sensors/instructions), which specify that the highlights should be submitted in the Front Matter section—together with the Title, Author list, Affiliations, Abstract, and Keywords.
Comments 2: Mention the main quantitative results in the abstract. Response 2: Due to the volume of the results, which are distributed across 8 figures and 3 tables, we faced challenges in condensing them into a few words for the abstract. Instead, we opted to provide the reader with a general overview to help guide their understanding of the findings. We hope the reviewer can understand our choice.
Comments 3: Add hyperlinks to references, tables, and figures. Response 3: Thank you for your comment. We understand the importance of including hyperlinks to references, tables, and figures. However, we assumed that these hyperlinks would be automatically inserted during the journal’s typesetting and final formatting process.
Comments 4: At the end of the introduction, add a paragraph describing the manuscript's structure. Add a brief description between section and subsection titles. Response 4: Thank you for your suggestion. To address your request, we have outlined the main steps of the manuscript’s structure. The updated version of this description can be found at the end of the introduction, as shown below: The manuscript is organized to guide the reader through the development and analysis of the study. It begins with a description of the construction and development of the weighing balance, followed by an explanation of the load cell arrangement and signal acquisition system. Next, we discuss how raw data is transformed into weight values through calibration and outline the experimental setup, including the rodents and SPA recordings. The methodology section continues with an introduction to the algorithm used to analyze SPA and the mathematical strategies applied, including Biesiadecki’s summation method and the Mean of Weight Changes (MWC) approach. We also explore the concept of signal dispersion and its role in SPA analysis. The classification of SPA into distinct domains is explained, along with the creation of heat maps for minute-by-minute SPA domains. In the results section, we compare different mathematical strategies, present an overview of the 23-day experiment, and examine analyses across light and dark phases, as well as across different times of the day. Further analyses explore the SPA classification into domains both within hours and across all experimental days.
Comments 5: In Figures 1 and 2, decimal division is indicated by a comma; please correct it. Response 5: Thank you for pointing this out. We have corrected the decimal separators in Figures 1 and 2 by replacing the commas with dots, in accordance with the standard formatting conventions.
Comments 6: The resolution of the figures needs to be improved. Response 6: We have updated all figures to higher-resolution versions to enhance visual quality and ensure that all details are clearly visible. Additionally, we have also provided the original figures as separate files in the submission system for further reference.
Comments 7: Please define all variables and parameters in the equations and figures in the main text. Response 7: We have made every effort to define all variables and parameters in the equations. However, if the reviewer identifies any specific points that remain unclear, we are more than happy to make further improvements and provide clarification as needed.
Comments 8: The article does not include any calculations or reasoning to justify why 10 animals per group (n = 20) are sufficient for the statistical analyses performed. "Newly weaned mice" are mentioned, but the inclusion criteria are not clearly defined, nor is it reported whether animals were discarded or for what reasons. A better description of the selected animals is needed. Response 8: The reviewer raised an important concern, and a robust response is necessary to meet their expectations. Indeed, in this study, we did not perform a sample size estimation to test our hypotheses. This study is part of a larger research project investigating various outcomes, including the impact of housing space on physiological markers and the deleterious effects of obesity. Not all outcomes are directly related to the present analysis. Specifically, to validate the use of the apparatus developed for the broader project, we initiated the study with control animals to assess the reliability of the equipment in monitoring SPA. Moreover, in accordance with the principles of the 3Rs and the ethical and appropriate use of laboratory animals, the Institutional Animal Care and Use Committee approved the proposed sample size based on previous studies conducted by our research group (Scariot, Manchado-Gobatto et al. 2019, Polisel, Beck et al. 2021, Scariot, Manchado-Gobatto et al. 2021, Scariot, Gobatto et al. 2022, Scariot, Manchado-Gobatto et al. 2022). These prior studies demonstrated satisfactory results related to SPA in rodent models and served as the scientific foundation for the current report. From this perspective, we opted to use a sample size consistent with our earlier work, which had shown high statistical power and methodological reliability. Nonetheless, we fully acknowledge the reviewer's concern and recognize this point as a limitation of our study. A corresponding statement has been added to the discussion section.
Comments 9: The density per animal in the cages varies between groups. This introduces an additional variable that is not controlled separately from cage size. Response 9: Thank you for your insightful comment. We intentionally designed the study so that the floor area per animal would differ between the large cage (267.9 cm²) and the small cage (49.5 cm²) to allow more space for movement in the larger cage. Our goal was to stimulate SPA by providing animals in the large cage with an environment that more closely resembles natural conditions, where rodents have the freedom to explore and move. In contrast, limited space in smaller cages imposes a sedentary lifestyle, which can lead to physiological adaptations that differ markedly from those observed in animals given greater environmental space. We chose not to equalize housing density across cages because doing so would have introduced other confounding issues. To make it easier to understand, let’s say that in the small cage, each mouse has 49.5 cm² of space. If we wanted to keep the same amount of space per mouse in the large cage, we would have to place 54 mice inside it, even though the large cage is much bigger. This would equalize the space per animal, but it would introduce a severe imbalance between the groups (10 mice in the small cage vs. 54 mice in the large cage) and create confounding effects related to social density and overcrowding. Therefore, for these reasons, we preferred to fix the number of animals at 10 per cage in both conditions to keep the number of social interactions relatively constant between groups. We believe our approach was the best option to provide sufficient space for stimulating SPA without introducing social order issues.
Comments 10: The proposed system is not compared to another reference method, which weakens claims about its validity. The system is inferred to be "robust and valid." However, validation is limited to detecting expected differences between large and small cages or between light/dark phases, representing a circular analysis. Response 10: Thank you to the reviewer for this insightful comment. Comparing the system to an another reference method would indeed strengthen claims about its validity, and this remains an important goal for future work. However, we respectfully clarify that our validation approach is not circular. In circular analyses, hypotheses are typically derived from the system’s output itself, which would result in the system being validated by the very data it generates. In contrast, our hypotheses were based on well-established theoretical backgrounds and findings from the literature, not derived from the system’s data. Specifically, we predicted that the system would detect two known patterns of SPA: (1) increased activity during the dark phase and (2) higher activity in larger cages. By confirming these patterns, which are already expected based on the literature and biological principles, we can assert that our balance demonstrates construct validity.
Comments 11: No confidence intervals or effect sizes are reported (except once with Cohen's d). This prevents assessment of the true magnitude of the differences. Response 11: We understand the reviewer’s concern and agree that presenting confidence intervals is an important approach. However, considering the current layout and number of tables, we would like to inquire which specific results the reviewer considers most appropriate for the inclusion of this information. As our study encompasses topics from different fields, one of the challenges we aimed to address was ensuring the accessibility of the methods and results to professionals from various disciplines. To this end, careful attention was given to the presentation of the study. We believe that including confidence intervals in all applicable tables might detract some readers from the main findings. Therefore, we respectfully ask the reviewer to indicate which results would benefit most from the inclusion of confidence intervals.
Comments 12: There is no evidence of verification of compliance with the assumptions of homogeneity of variances or discussion of possible additional transformations beyond the inverse normalization. Response 12: We appreciate the reviewer’s comment regarding the absence of a formal assessment of variance homogeneity. This is indeed a valuable consideration when evaluating the robustness of statistical models. In our case, we chose not to apply standard tests of homogeneity (such as Levene’s or Bartlett’s), as we felt that the assumption of equal variances across groups was less central within the context of our analytical approach. First, our data were subjected to normal score transformation (also known as inverse normal transformation), a robust approach specifically used to address issues related to non-normal distributions. Transformation procedures are known for standardizing the distribution by mapping the data to a normal score scale, thereby stabilizing the variance (i.e., minimizing heteroscedasticity) (Kendler, Gardner et al. 2011, Roux, Fraïsse et al. 2014). Moreover, ANOVA remains relatively robust to violations of homoscedasticity (Lix, Keselman et al. 1996), especially when group sizes are equal or nearly equal, as was the case in our experimental design. Therefore, considering (1) the statistical robustness provided by the transformation, (2) the balanced sample sizes across groups, and (3) the exploratory nature of the biological data involved—we respectfully opted not to include formal homogeneity tests. We are grateful for the reviewer’s observation, which has allowed us to discuss in a high level the rationale behind our analytical choices.
Comments 13: Table 2 lacks statistical analysis due to non-normality. Nonparametric methods could have been used for at least an exploratory comparison between hours. Response 13: Upon revisiting the analysis and exploring alternative procedures, we aimed to identify a test capable of evaluating the two main effects in our factorial design. However, we found that the Kruskal–Wallis test was not suitable, as it is inherently limited to one-way comparisons (Chan and Walmsley 1997). Thus, we found the rank transformation to be the most accessible alternative (Conover and Iman 1981). Although not a purely nonparametric test, this approach is useful for data that violate parametric assumptions. To implement it, the entire set of observations is ranked from smallest to largest—assigning rank 1 to the smallest value, rank 2 to the next smallest, and so on up to the maximum value, which receives the highest rank. The ranked data can then be analyzed using standard parametric methods (for example, a two‑way factorial ANOVA). This enables us to evaluate both the effect of housing and the effect of domain simultaneously.
Comments 14: How signals are synchronized or how precise temporal alignment between cells is ensured is not specified. The conversion of raw signal to weight units is explained, but the calibration standard errors are not detailed. Response 14: Data synchronization between scales was not an issue, as each raw data sample was recorded with a timestamp accurate to the microsecond. We have now provided a more detailed explanation of the estimation error and included the paragraph below to address the reviewer's comments. The Standard Error of the Estimate (SEE) was calculated using the calibration data from the load cells. It was determined by taking the root mean value of the residual deviations between the observed and predicted measurements, adjusted for the degrees of freedom. The analysis yielded an SEE of approximately 0.4 grams across the three load cells, indicating that the predicted masses deviate from the actual known values by only 0.4 grams. This low level of error confirms that the system has satisfactory sensitivity to detect subtle weight variations in small animal models.
Comments 15: Variability in sampling frequency (unstable 40 Hz). If not properly controlled, this can introduce artifacts. MWC mitigates this, but its impact on longitudinal studies is not analyzed. |
Response 15: We acknowledge that differences in sampling frequencies can primarily affect the SWC, and we appreciate this insightful comment. All balances were built using the same hardware, resulting in similar sampling frequencies; thus, differences between systems should be minimal. However, some variation both between and within systems is expected. To improve the accuracy and precision of the systems, we adopted the following strategy: We assumed that the minute with the lowest SPA within a day reflects the time when the animals exhibited minimal movement. Therefore, the SPA recorded during that minute likely represents the system's baseline bias for that day. Consequently, we subtracted the value of this minimum activity from all minute-by-minute SPA values recorded on the same day. This correction enhances the accuracy of each system and reduces discrepancies between systems across the study period. Furthermore, it improves the consistency of measurements within each system across different days. We have included these methodological details in the manuscript to strengthen the technical rigor of the data analysis section.
|
|
|
|
|
Chan, Y. and R. P. Walmsley (1997). "Learning and understanding the Kruskal-Wallis one-way analysis-of-variance-by-ranks test for differences among three or more independent groups." Phys Ther 77(12): 1755-1762.
Conover, W. J. and R. L. Iman (1981). "Rank transformations as a bridge between parametric and nonparametric statistics." The American Statistician 35(3): 124-129.
Kendler, K. S., C. Gardner and D. M. Dick (2011). "Predicting alcohol consumption in adolescence from alcohol-specific and general externalizing genetic risk factors, key environmental exposures and their interaction." Psychological Medicine 41(7): 1507-1516.
Lix, L. M., J. C. Keselman and H. J. Keselman (1996). "Consequences of Assumption Violations Revisited: A Quantitative Review of Alternatives to the One-Way Analysis of Variance "F" Test." Review of Educational Research 66(4): 579-619.
Polisel, E. E. C., W. R. Beck, P. P. M. Scariot, T. M. M. Pejon, C. A. Gobatto and F. B. Manchado-Gobatto (2021). "Effects of high-intensity interval training in more or less active mice on biomechanical, biophysical and biochemical bone parameters." Scientific Reports 11(1): 6414.
Roux, C., C. Fraïsse, V. Castric, X. Vekemans, G. H. Pogson and N. Bierne (2014). "Can we continue to neglect genomic variation in introgression rates when inferring the history of speciation? A case study in a Mytilus hybrid zone." Journal of Evolutionary Biology 27(8): 1662-1675.
Scariot, P. P. M., C. A. Gobatto, E. E. C. Polisel, A. E. C. Gomes, W. R. Beck and F. B. Manchado-Gobatto (2022). "Early-life mice housed in standard stocking density reduce the spontaneous physical activity and increase visceral fat deposition before reaching adulthood." Lab Anim 56(4): 344-355.
Scariot, P. P. M., F. B. Manchado-Gobatto, W. R. Beck, M. Papoti, P. R. Van Ginkel and C. A. Gobatto (2022). "Monocarboxylate transporters (MCTs) in skeletal muscle and hypothalamus of less or more physically active mice exposed to aerobic training." Life Sciences 307: 120872.
Scariot, P. P. M., F. B. Manchado-Gobatto, T. A. Prolla, I. G. Masselli Dos Reis and C. A. Gobatto (2019). "Housing conditions modulate spontaneous physical activity, feeding behavior, aerobic running capacity and adiposity in C57BL/6J mice." Hormones and Behavior 115: 104556.
Scariot, P. P. M., F. B. Manchado-Gobatto, P. R. Van Ginkel, T. A. Prolla and C. A. Gobatto (2021). "Aerobic training associated with an active lifestyle exerts a protective effect against oxidative damage in hypothalamus and liver: The involvement of energy metabolism." Brain Research Bulletin 175: 116-129.

Round 2
Reviewer 3 Report
Comments and Suggestions for Authors
My comments have been addressed. Thank you.